# A trimeric Rab7 GEF controls NPC1-dependent lysosomal cholesterol export

Dick J. H. van den Boomen [1✉], Agata Sienkiewicz[1], Ilana Berlin[2], Marlieke L. M. Jongsma[2], Daphne M. van Elsland[2], J. Paul Luzio [3], Jacques J. C. Neefjes [2] & Paul J. Lehner [1✉]

Cholesterol import in mammalian cells is mediated by the LDL receptor pathway. Here, we perform a genome-wide CRISPR screen using an endogenous cholesterol reporter and identify >100 genes involved in LDL-cholesterol import. We characterise C18orf8 as a core subunit of the mammalian Mon1-Ccz1 guanidine exchange factor (GEF) for Rab7, required for complex stability and function. C18orf8-deficient cells lack Rab7 activation and show severe defects in late endosome morphology and endosomal LDL trafficking, resulting in cellular cholesterol deficiency. Unexpectedly, free cholesterol accumulates within swollen lysosomes, suggesting a critical defect in lysosomal cholesterol export. We find that active Rab7 interacts with the NPC1 cholesterol transporter and licenses lysosomal cholesterol export. This process is abolished in C18orf8-, Ccz1- and Mon1A/B-deficient cells and restored by a constitutively active Rab7. The trimeric Mon1-Ccz1-C18orf8 (MCC) GEF therefore plays a central role in cellular cholesterol homeostasis coordinating Rab7 activation, endosomal LDL trafficking and NPC1-dependent lysosomal cholesterol export.

[1] Cambridge Institute of Therapeutic Immunology & Infectious Disease, University of Cambridge, Cambridge, UK. [2] Leiden University Medical Centre, Leiden University, Leiden, The Netherlands. [3] Cambridge Institute for Medical Research, University of Cambridge, Cambridge, UK. ✉email: djhv2@cam.ac.uk; pjl30@cam.ac.uk

C holesterol is an essential component of all mammalian lipid membranes and disturbances in its homeostasis associate with a variety of diseases, including Niemann Pick type C, familial hypercholesterolemia, atherosclerosis and obesity. Cellular cholesterol homeostasis is transcriptionally regulated by the SREBP2 (sterol response element binding protein 2) transcription factor. When cholesterol is abundant SREBP2 and its trafficking factor SCAP are retained in the endoplasmic reticulum (ER), whereas upon sterol depletion they are released to the Golgi, where SREBP2 is cleaved and traffics to the nucleus to activate the transcription of genes involved in de novo cholesterol biosynthesis (e.g. *HMGCS1*, *HMGCR*, *SQLE*) and LDL-cholesterol uptake (e.g. *LDLR*, *NPC1/2*)[1]. SREBP2 thus couples cholesterol import and biosynthesis to cellular cholesterol availability.

Cholesterol is taken up into mammalian cells in the form of low-density lipoprotein (LDL) particles, composed predominantly of cholesterol esters (CEs), cholesterol and apolipoprotein B-100[2]. LDL binds to the cell surface LDL receptor (LDLR) and the LDLR-LDL complex is internalised via clathrin-mediated endocytosis[3]. In the acidic pH of sorting endosomes, LDL dissociates from LDLR, which recycles back to the plasma membrane. LDL is itself transported to a late endosomal/lysosomal (LE/Ly) compartment, where cholesterol is released from CEs by lysosomal acid lipase (LAL/LIPA). Free cholesterol then binds the Niemann Pick C2 (NPC2) carrier protein and is transferred to the Niemann Pick C1 (NPC1) transporter for subsequent export to other organelles[4].

NPC1 is a central mediator of lysosomal cholesterol export. Mutations in *NPC1* and *NPC2* cause Niemann Pick type C, a lethal lysosomal storage disease characterised by lysosomal cholesterol accumulation in multiple organ systems. *NPC1* encodes a complex polytopic glycoprotein embedded in the lysosomal membrane[5]. Cholesterol-loaded NPC2 binds the NPC1 middle luminal domain[6,7] and delivers its cargo to the mobile N-terminal domain[8,9], from where cholesterol is likely transferred to the sterol-sensing domain (SSD). Whether cholesterol is transported through NPC1 in a channel-like manner or simply inserted into the luminal membrane leaflet remains unclear[10]. NPC1 is thought to be constitutively active and no regulators of its activity have been identified. Once transported across the lysosomal membrane, cholesterol transfer to other organelles is mediated by lipid transfer proteins (LTPs) at inter-organelle membrane contact sites (MCS)[11]. Although a variety of proteins have been implicated in MCS formation with LE/Ly[12–15], the identity of the direct carrier transporting cholesterol from LE/Ly remains controversial.

The LE/Ly compartment plays a central role in cellular LDL-cholesterol uptake. LE homeostasis and substrate trafficking are regulated by the Rab7 GTPase[16]. Rab7 activity is controlled by its nucleotide status: its activation requires GDP-to-GTP exchange by a guanidine exchange factor (GEF), whereas GTP hydrolysis induced by GTPase activating proteins (GAPs) triggers Rab7 inactivation. The active GTP-bound Rab7 associates with LE membranes and recruits Rab7 effector proteins that mediate a variety of effector functions including endosome-to-lysosome fusion, endosome motility and recycling[16]. In mammalian cells Rab7 effectors include the motility factor RILP[17,18], the cholesterol binding protein ORP1L[19,20], the VPS34 phosphatidylinositol (PtdIns) 3-kinase regulators Rubicon and WDR91[21,22] and the retromer components VPS26 and VPS35[23,24].

Temporal and spatial control of Rab7 activation is of major importance. At least three GAPs – TBC1D5, TBC1D15 and Armus[24–26] are implicated in Rab7 inactivation, whereas activation of the yeast Rab7 homologue Ypt7 is mediated by the Mon1-Ccz1 (MC1) complex[27,28]. MC1 is recruited to endosomal membranes by the phospholipid PtdIns(3)P[29] and its activation

of Rab7 drives Rab5-to-Rab7 conversion, endosome maturation and fusion with the vacuolar/lysosomal compartment[30–33]. A partial crystal structure of the *C. thermophilum* MC1-Ypt7 complex suggests a GEF model in which MC1 binding to Ypt7 induces magnesium expulsion and GDP dissociation from the Ypt7 active site[34]. Mammals have two Mon1 orthologues— Mon1A and Mon1B—and a single Ccz1 orthologue (encoded by two near-identical genes *Ccz1* and *Ccz1B*). Although mammalian MC1 also likely acts as a Rab7 GEF[35,36], its in vivo function remains poorly characterised.

In this study, we perform a genome-wide and sub-genomic CRISPR screen for essential genes in cholesterol homeostasis using an endogenous SREBP2-dependent cholesterol reporter. We identify >100 genes mainly involved in the LDL-cholesterol uptake pathway, including the poorly characterised *C18orf8*. C18orf8 is a novel core component of the mammalian MC1 complex, essential for complex stability and function. *C18orf8*-deficient cells exhibit severe defects in Rab7 activation and LDL trafficking, concomitant with swelling of the LE/Ly compartment and marked lysosomal cholesterol accumulation. We show that active Rab7 interacts with the NPC1 cholesterol transporter to license lysosomal cholesterol export - a pathway which is deficient in *C18orf8*-, *Ccz1*- and *Mon1A/B*-deficient cells and restored by a constitutively active Rab7 (Q67L). Our findings therefore identify a central role for the trimeric Mon1-Ccz1-C18orf8 (MCC) GEF in cellular LDL-cholesterol uptake, coordinating Rab7 activation with LDL trafficking and NPC1-dependent lysosomal cholesterol export.

## Results

### Generation of an endogenous, SREBP2-dependent, fluorescent cholesterol reporter.

To screen for genes that maintain cellular cholesterol homeostasis, we engineered a cell line expressing an endogenous fluorescent reporter responsive to intracellular cholesterol levels. HMG-CoA synthase 1 (HMGCS1) catalyses the second step of cholesterol biosynthesis, the conversion of acetoacetyl-CoA and acetyl-CoA to HMG-CoA. With two sterol-response elements (SRE) in its promoter, HMGCS1 expression is highly SREBP2 responsive and thus cholesterol sensitive[37], rendering it well suited to monitor cellular cholesterol levels (Fig. 1a). We used CRISPR technology to knock-in the bright fluorescent protein Clover[38] into the endogenous *HMGCS1* locus yielding an endogenous HMGCS1-Clover fusion protein (Fig. 1b and Supplementary Fig. 1a). Sterol depletion of the *HMGCS1-Clover* cell line, using mevastatin and lipoprotein-depleted serum (LPDS), increased basal HMGCS1-Clover expression 9-fold (Fig. 1d and Supplementary Fig. 1b, c) and revealed the expected cytoplasmic localisation of the HMGCS1-Clover fusion protein (Fig. 1c). Expression returned to baseline following overnight sterol repletion (Supplementary Fig. 1d). Sterol-depletion induced HMGCS1-Clover expression was abolished upon knockout of *SREBP2*, but not its close relative *SREBP1* (Fig. 1d and Supplementary Fig. 1e). *SREBP2*-deficiency also slightly decreased steady-state HMGCS1-Clover levels (Supplementary Fig. 1f), suggesting low-level SREBP2 activation under standard growth conditions. Unlike HMG-CoA reductase (HMGCR), HMGCS1 expression is not regulated by sterol-induced degradation (Supplementary Fig. 1g). *HMGCS1-Clover* is, therefore, a sensitive endogenous, SREBP2-dependent, transcriptional reporter for intracellular cholesterol levels.

### A genome-wide CRISPR screen identifies components essential for cellular cholesterol homeostasis.

To identify genes critical to the maintenance of cellular cholesterol levels, we undertook a genome-wide CRISPR screen using our HMGCS1-Clover reporter as an intracellular cholesterol sensor (Fig. 1e). CRISPR knockout of

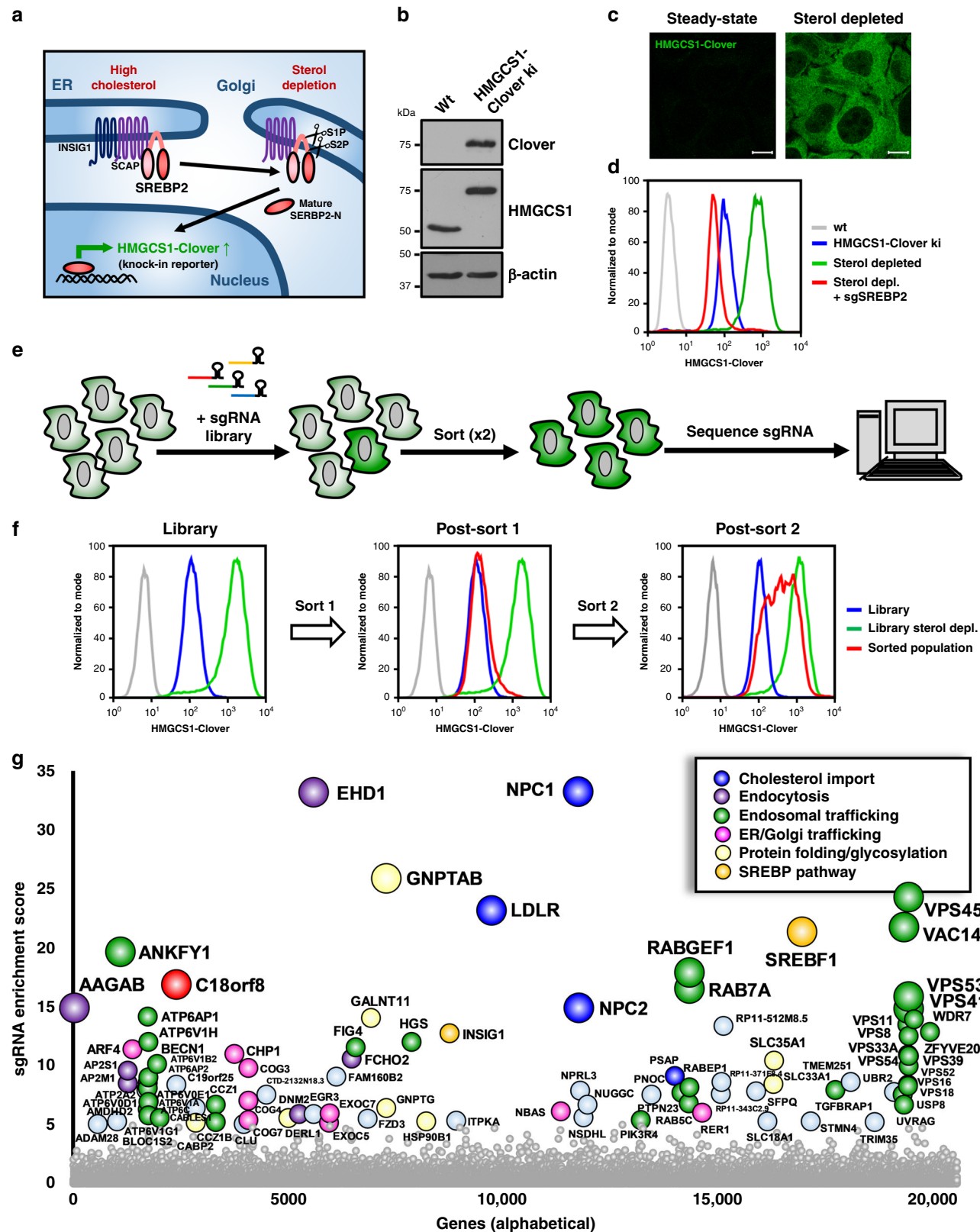

these genes is predicted to induce spontaneous cholesterol depletion, SREBP2 activation and therefore HMGCS1-Clover upregulation, despite sterol replete (high cholesterol) culture conditions. These cells thus develop a spontaneous HMGCS1-Clover[high] phenotype. *HMGCS1-Clover* reporter cells were targeted with a genome-wide CRISPR library containing 10 sgRNAs per gene (total

sgRNA size 220,000)[39] and were sorted in two successive rounds of flow cytometry for the rare HMGCS1-Clover[high] phenotype (Fig. 1e). This yielded an enriched population with a 60% HMGCS1-Clover[high] phenotype under sterol replete conditions (Fig. 1f). Next-generation sequencing of the population revealed sgRNA enrichment for 93 genes (MAGeCK sgRNA enrichment

**Fig. 1 A genome-wide CRISPR screen identifies essential factors in cellular cholesterol homeostasis. a**, **b** Creation of a HMGCS1-Clover CRISPR knock-in reporter in HeLa cells. **a** Schematic representation of the HMGCS1-Clover knock-in reporter and its induction upon sterol depletion by the SREBP2 transcription factor. **b** Immunoblotting of a reporter clone shows the endogenous HMGCS1-Clover fusion protein, detected by both HMGCS1- and GFP/Clover-specific antibody staining. **c**, **d** Sterol depletion induces HMGCS1-Clover expression in an SREBP-dependent manner. **c** HeLa HMGCS1-Clover cells were sterol depleted overnight and HMGCS1-Clover expression was analysed by microscopy. Representative images are shown from five fields per condition. Scale bars = 10 μm. **d** HeLa HMGCS1-Clover CAS9 cells were transfected with sgRNA against *SREBF2* (red line) or control (blue and green lines), sterol depleted at day 7 (green and red lines) and HMGCS1-Clover expression was determined by flow cytometry. **e**–**g** Genome-wide CRISPR screening using the HMGCS1-Clover cholesterol reporter. **e** Schematic for the identification of genes essential for cellular cholesterol homeostasis. **f** HeLa HMGCS1-Clover/Cas9 cells were transduced with a genome-wide sgRNA library (220,000 sgRNAs). Rare HMGCS1- Clover[high] cells were isolated using two rounds of cell sorting, resulting in a 60% enriched HMGCS1-Clover[high] population (right panel, red line). **g** Illumina sequencing of sgRNAs in the isolated HMGCS1-Clover[high] population shows sgRNA enrichment for genes involved in cholesterol uptake (LDLR, NPC1, NPC2; blue), protein folding and glycosylation (yellow), membrane trafficking of LDLR/LDL (pink, purple, green) and SREBP2 function (orange). Genes with MAGeCK sgRNA enrichment score < $10^{-5}$ are indicated by enrichment score and gene name. The full dataset is available in Supplementary Data 1.

score <$10^{-5}$) (Fig. 1g, Supplementary Data 1). The results of our HMGCS1-Clover screen emphasise the central role of LDL-cholesterol uptake and membrane trafficking in cellular cholesterol homeostasis (Fig. 1g, Fig. 2c), with top hits including *LDLR* and *NPC1/2*.

To validate our genome-wide screening results, we took the top 1000 ranking genes from the genome-wide screen and created a sub-genomic sgRNA library containing 10 sgRNAs per gene and 2500 non-targeting sgRNAs (Fig. 2a; total sgRNA size 12,340, Supplementary Data 1). Screening this library as above yielded an enriched population with a 75% HMGCS1-Clover[high] phenotype (Supplementary Fig. 2a) and sgRNA enrichment for 65 genes (MAGeCK sgRNA enrichment score <$10^{-4}$) (Supplementary Fig. 2b, Supplementary Data 1). The majority of these hits (53) overlapped with hits from our genome-wide screen (Fig. 2a, b, red), confirming reproducibility of our approach. In all, 41 hits were unique to the genome-wide screen (blue) and 13 were unique to our targeted sub-genomic screening approach (green). From the identity of these unique hits, it is clear that the two screening approaches are complementary as well as over-lapping. Whereas the genome-wide screen, for example, identified multiple components of the COG (COG3/4/7) and vacuolar ATPase (ATP6AP1/AP2/V0D1/V0E1/V1A/V1B2/V1G1/V1H) complexes, the targeted screen identified additional subunits (COG5/8, ATP5V1D/V1F) as well as the three V-ATPase assembly factors CCDC115, TMEM199 and VMA21 (Fig. 2b). Conversely, whereas both screens identified the AP2 μ1 subunit (AP2M1), the genome-wide screen additionally identified the AP2 σ1 subunit (AP2S1), the AP2 assembly factor AAGAB and accessory factor FCHO2. Our genome-wide and targeted screening approaches thus provide an extensive overlapping dataset of 106 genes required for cellular cholesterol homeostasis (Fig. 2a, b and Supplementary Data 1).

Functionally, the hits from our two screens highlight the importance of LDL-cholesterol import and membrane trafficking in cellular cholesterol homeostasis. They can be grouped in functional categories (Fig. 1g and Supplementary Fig. 2b) that include: (i) protein folding and glycosylation (yellow, e.g. *HSP90B1, GNTAP, SLC35A1*); (ii) early secretory pathway (pink, e.g. *ARF1/4, COPB1/B2/G2, CHP1, COG3/4/5/7/8, EXOC5/7*); (iii) endocytosis (purple, e.g. *AP2M1/S1, AAGAB, EHD1, FCHO2, DNM2*); (iv) endosomal trafficking (green, e.g. *RabGEF1, CCZ1/1B, Rab5C, Rab7A*, V-ATPase (*ATP6AP1/AP2/V0D1/V0E1/V1A/V1B2/V1D/V1F/V1G/V1H*), ESCRT (*PTPN23, HRS*), CORVET (*VPS8/16/18/TGFBRAP1*), HOPS (*VPS16/18/39/41*), FERARI (*ZFYVE20, VPS45*), PI3K (*BECN, UVRAG, PIK3R4/C3*), PIKfyve (*FIG4, VAC14*)); and v) cholesterol import (blue, *LDLR, NPC1/2*). These categories reflect successive stages in LDL-cholesterol import (Fig. 2c), respectively: (i) LDLR folding and glycosylation; (ii) trafficking of LDLR to the cell surface; (iii) endocytosis of the

LDL-LDLR complex; (iv) trafficking of internalised LDL to the LE/Ly compartment; and v) lysosomal cholesterol release and NPC1-dependent lysosomal cholesterol export.

Besides the LDLR pathway, our screen identified components of the SREBP transcriptional machinery (Fig. 1g and Supplementary Fig. 2b, orange, *INSIG1, SREBP1*) and several poorly characterised gene products (grey/red e.g. *CLU/APOJ, EGR3, ADAM28, FAM160B2, STMN4, C18orf8*). To validate our screening results, we confirmed a subset of hits (*LDLR, NPC1, AP2μ1, ANKFY1, VPS16, ZFYVE20, INSIG1, SREBF1*) using individual sgRNAs. All sgRNA-treated cells showed elevated HMGCS1-Clover expression (Supplementary Fig. 3), suggesting defective LDL-cholesterol import and/or spontaneous SREBP2 activation.

Our cholesterol reporter screens, therefore, provide an extensive overview and unique insight into both known and unknown factors essential for mammalian LDL-cholesterol import (Figs. 1g, 2c and Supplementary Fig. 2b and Supplementary Data 1), and emphasise the critical role of endo-/lysosomal trafficking within this process.

**C18orf8 is required for endo-lysosomal LDL-cholesterol uptake.** A prominent uncharacterised gene in our screens is *C18orf8* (Fig. 1g and Supplementary Fig. 2b, red), which encodes a 72 kDa soluble protein, conserved from animals to plants with no identifiable yeast orthologue. Structure prediction (Phyre2, HHPred[40,41]) suggests a high structural similarity to the N-terminus of clathrin heavy chain, with an N-terminal β-propeller composed of WD40 repeats and a C-terminal α-helical domain (Supplementary Fig. 5d). Organelle proteomics has previously identified C18orf8 as a lysosome[42] and endosome[43] associated protein.

To further characterise *C18orf8* function, we generated *C18orf8* knockout clones using two independent sgRNAs (Fig. 3a). Consistent with our screening results, *C18orf8*-deficient cell clones showed elevated HMGCS1-Clover expression under sterol-replete culture conditions (Fig. 3b, red line) – implying spontaneous cholesterol depletion – and this phenotype was reversed upon re-expression of HA-tagged *C18orf8* (C18orf8-3HA, green line). Cholesterol deficiency could result from defective exogenous LDL-cholesterol uptake or defective endogenous biosynthesis. To differentiate these pathways, we inhibited exogenous uptake by culturing cells in LPDS or blocked endogenous biosynthesis with mevastatin (Fig. 3c). Wild-type reporter HeLa cells showed no response to mevastatin alone (Fig. 3c, left panel, red line), upregulated HMGCS1-Clover in LPDS (green line) and showed maximum reporter induction in LPDS and mevastatin (blue line). In contrast, *C18orf8*-deficient cells showed elevated steady-state HMGCS1-Clover expression (Fig. 3c, middle/right panels, black lines), were unresponsive to LPDS (green lines), but addition of mevastatin alone induced maximum HMGCS1-Clover induction (red lines) with no further

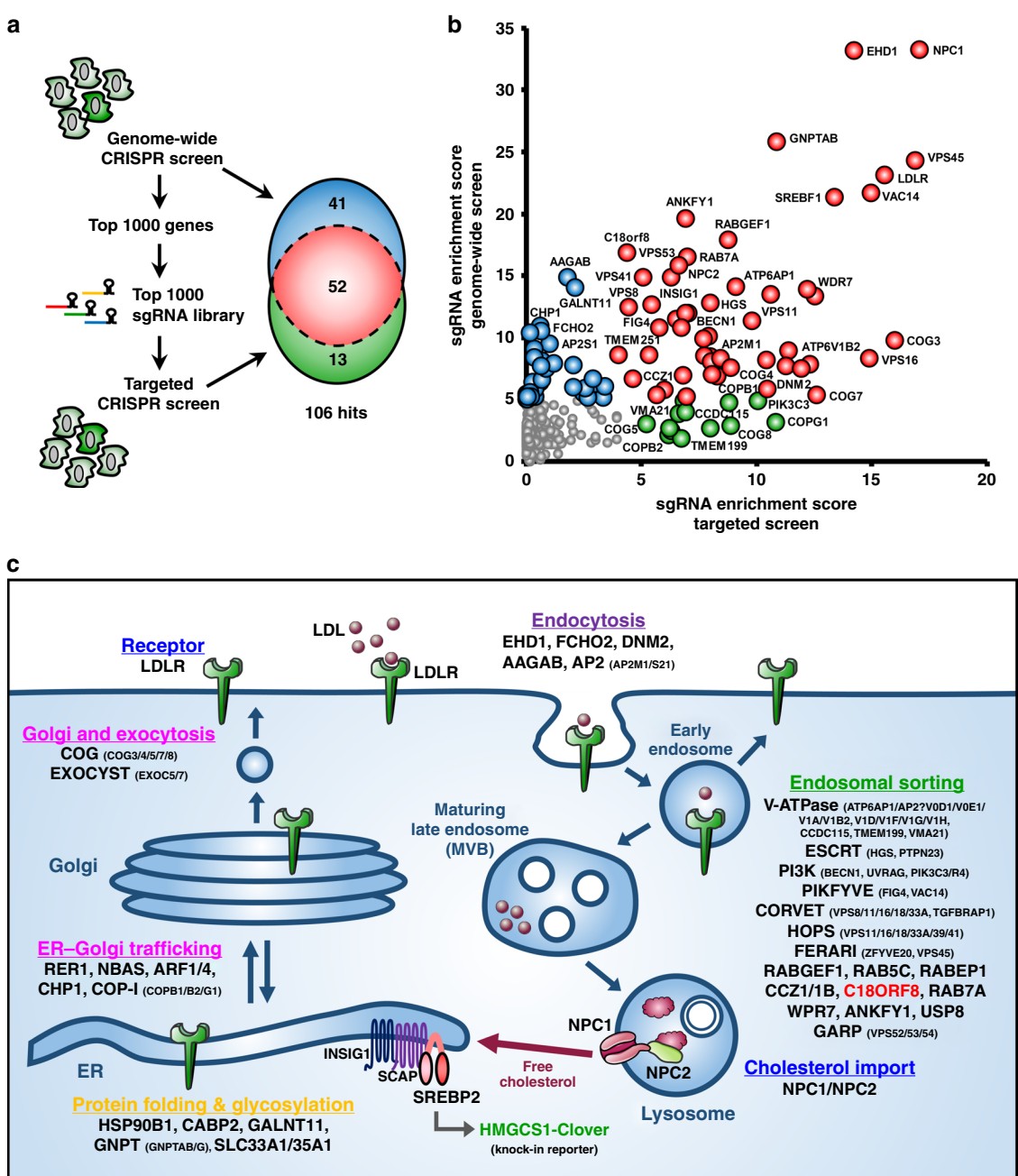

**Fig. 2 A secondary sub-genomic CRISPR screen validates genes required for cellular LDL-cholesterol uptake. a** Schematic of the primary (genome-wide) and secondary (targeted, top 1000) screening approach. Screening results from both screens show extensive overlap and create a complementary dataset of 106 genes required for cellular cholesterol homeostasis. **b** Comparison of the genome-wide (vertical) and targeted (horizontal) CRISPR screening results. Genes are indicated by MAGeCK sgRNA enrichment score, with hits specific to the genome-wide screen (score < 10$^{-5}$) indicated in blue, hits specific to the targeted screen (score < 10$^{-4}$) in green and hits in both screens in red. A select number of hits is indicated by gene name. The full datasets are available in Supplementary Fig. 2 and Supplementary Data 1. **c** A schematic representation of combined hits from the genome-wide and targeted CRISPR screens highlights the central role of LDL-cholesterol import in cellular cholesterol homeostasis. Hits are grouped by membrane trafficking pathway of their involvement. A full list of hits from both screens is available in Supplementary Data 1.

increase in LPDS plus mevastatin (blue line). These results show that while wild-type HeLa cells rely predominantly on external cholesterol and switch to endogenous biosynthesis when LDL is unavailable, *C18orf8*-deficient cells are completely reliant on endogenous cholesterol biosynthesis under all conditions. This effect is likely explained by a defect in LDL-cholesterol uptake.

*C18orf8*-deficient cells showed normal LDLR cell surface expression (Supplementary Fig. 4a) and uptake of fluorescent Dil-LDL (Fig. 3d). However, whereas in wild-type cells Dil-LDL

fluorescence disappeared within 3 h of pulse-labelling, in *C18orf8*-deficient cells Dil-LDL accumulated during the 3-h chase (Fig. 3e), a phenotype rescued upon complementation with C18orf8-3HA. To determine whether this defect is LDL-specific, or derives from a general endo-lysosomal trafficking defect, we stimulated cells with fluorescent-labelled EGF. Similar to Dil-LDL, EGF was degraded within 3 h in wild-type cells, but accumulated during the 3 h chase in *C18orf8*-deficient cells (Supplementary Fig. 4b). *C18orf8*-deficient cells thus show a general defect in

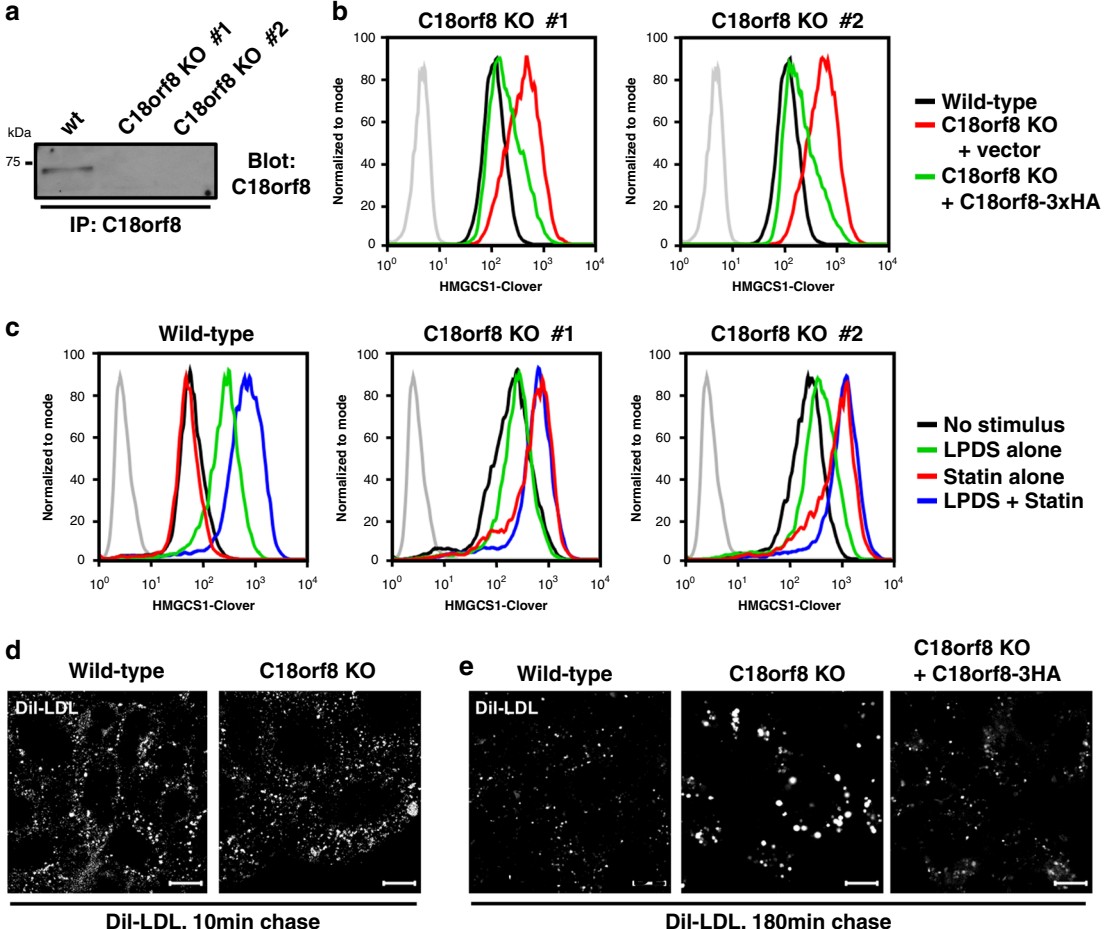

**Fig. 3 C18orf8 is required for endosomal LDL-cholesterol uptake. a**, **b** *C18orf8*-deficient cells show spontaneous cholesterol deficiency. **a** Wild-type and *C18orf8*-deficient clones were lysed, endogenous C18orf8 immunoprecipitated, and detected by a C18orf8-specific antibody. **b** *C18orf8*-deficient HMGCS1-Clover clones were transduced with HA-tagged *C18orf8* (C18orf8-3xHA; green lines) or empty vector (red lines) and HMGCS1-Clover expression was determined by flow cytometry at day 18 using wild-type HMGCS1-Clover cells as a control (black lines) (three independent experimental replicates). **c** *C18orf8*-deficient cells are dependent on endogenous cholesterol biosynthesis. Wild-type and *C18orf8*-deficient HMGCS1-Clover cells were either cultured in LPDS to block exogenous LDL-cholesterol uptake (green lines), treated with mevastatin to block endogenous cholesterol biosynthesis (red lines), or a combination of both treatments (blue lines), after which cells were analysed by flow cytometry for HMGCS1-Clover expression. **d**, **e** *C18orf8*-deficient cells show defective endo-lysosomal LDL degradation. Wild-type, *C18orf8*-deficient or *C18orf8*-deficient cells complemented with C18orf8-3xHA were starved for 1 h and pulse-labelled with fluorescent Dil-LDL, incubated for 5 min (**d**) or 180 min (**e**), fixed and visualised by confocal microscopy. Exposure times were kept constant between individual conditions at given time points. Representative images are shown from five fields per condition. Scale bars = 10 µm.

endo-lysosomal degradation that affects both LDL and EGF and results in their reliance on de novo cholesterol biosynthesis for cholesterol supply.

**C18orf8-deficient cells are defective in late endosome morphology and early-to-late endosomal trafficking.** Consistent with a defect in LDL/EGF degradation, *C18orf8*-deficient cells show severe disruption of endosome morphology, in particular of the LE/Ly compartment (Fig. 4a, b and Supplementary Fig. 4c). Perinuclear Rab7+ LEs (red) were markedly swollen and surrounded by equally swollen LAMP1+ LE/Lys (green), with EEA1+/Rab5+ EEs (blue) clustered between Rab7+ LEs in the perinuclear region. By electron microscopy (EM), the swollen LE/Lys appeared as enlarged multivesicular bodies (MVBs) with a high intraluminal vesicle (ILV) content and a 3-fold increase in average diameter (Supplementary Fig. 4d). Complementation of *C18orf8*-deficient cells with C18orf8xHA was slow, but after 14 days restored endosome morphology (Supplementary Fig. 4e) and function (Fig. 3e) back to wild-type. This prolonged recovery likely reflects the severity of the cellular phenotype.

To assess how this abnormal morphology affects substrate trafficking, we pulse-labelled wild-type and *C18orf8*-deficient cells with fluorescent-labelled EGF. In both wild-type and *C18orf8*-deficient cells EGF (red) moved to EEA1+ EE (blue) within ~20 min (Fig. 4c). However, whereas in wild-type cells EGF disappeared from EE at 1–3 h and was degraded (Fig. 4c, top panel), in *C18orf8*-deficient cells no degradation occurred and EGF remained in EEA1+ EE for the duration of the 3 h chase (bottom panel), implying a defect in early-to-late substrate trafficking. Indeed, inhibiting EGF degradation with Leupeptin, E-64d and Pepstatin, EGF (red) can be visualised to entry LAMP1+ LE/Ly (green) at 1–3 h in wild-type (Fig. 4d, top panel), but not *C18orf8*-deficient cells, in which co-localisation remained minimal throughout the chase (bottom panel). As seen with EGF, *C18orf8*-deficient cells also accumulated Dil-LDL in EEA1+ EE for the duration of a 3 h chase (Fig. 4e) and trafficking of the fluid phase dye Sulforhodamine 101 (SR101)[44] into lysotracker+ LE/Ly was delayed (Supplementary Fig. 4f). C18orf8, therefore, plays an essential role in early-to-late and/or late endosomal trafficking that precedes substrate degradation in lysosomes.

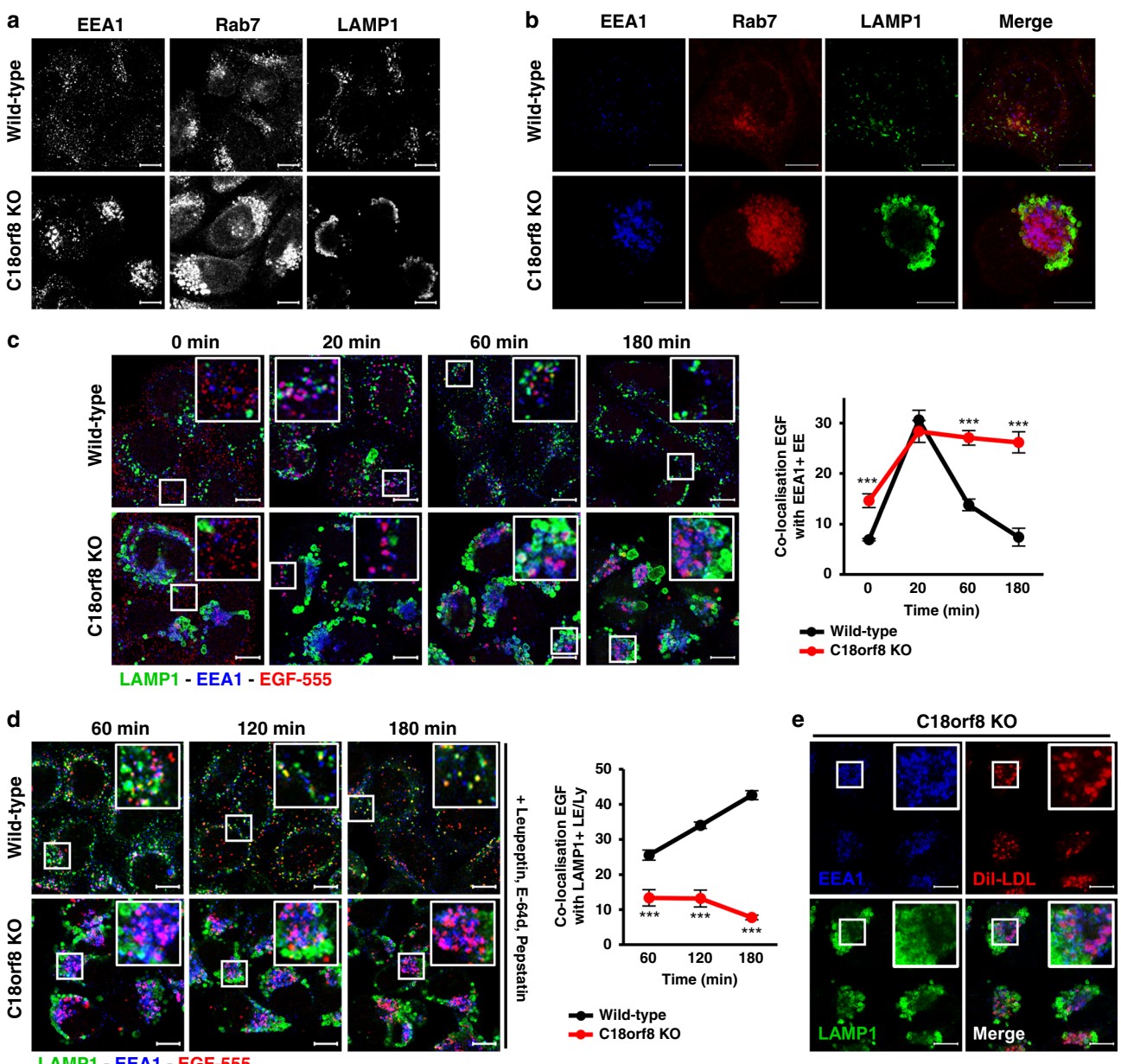

**Fig. 4 _C18orf8_-deficient cells show severe defects in late endosome morphology and early-to-late endosomal trafficking.** **a**, **b** _C18orf8_-deficient cells show clustering of EE and swelling of LE/Ly. Confocal microscopy comparing EEA1, Rab7 and LAMP1 single (**a**) or co-staining (**b**) in wild-type versus _C18orf8_-deficient cells. Representative images are shown from five fields per condition and 2 independent experiments. **c**–**e** _C18orf8_-deficient cells have severe defects in early-to-late endosomal trafficking. **c**, **d** Wild-type and _C18orf8_-deficient cells were starved for 1 h, pulse-labelled with AlexaFluor555-conjugated EGF (red), incubated for the indicated times, fixed and stained for EEA1 (blue) and LAMP1 (green). In **d** a cocktail of protease inhibitors (Leupeptin/E-64d/Pepstatin) was added during starve and chase to block EGF degradation. EEA1+ and LAMP1+ vesicles were identified using Volocity software and the percentage of EGF co-localising with these structures was determined from $n = 5$ fields per condition with at least eight cells per field. Error bars reflect standard error of mean (two-sided unpaired $t$-test, ***$p < 0.001$). **e** _C18orf8_-deficient cells were stimulated with DiI-LDL (red), chased for 3 h, fixed and stained for EEA1 (blue) and LAMP1 (green). Representative images are shown from five fields. Scale bars = 10 μm.

**C18orf8 is an integral component of the Mon1-Ccz1 complex.** To further characterise C18orf8 function, we identified C18orf8-interacting proteins by mass spectrometry. Pull-down of N- or C-terminal HA-tagged C18orf8 from HeLa cells revealed a dominant interaction with three proteins – Ccz1, Mon1A and Mon1B – the three components of the mammalian MC1 complex (Fig. 5a and Supplementary Data 2, PXD021444). The interaction between overexpressed C18orf8 and endogenous Ccz1 and Mon1B was readily confirmed by immunoblot (Fig. 5b). To visualise the interaction between endogenous proteins, we used CRISPR technology to knock-in a 3xMyc-tag into the C18orf8

locus, yielding an endogenous C18orf8-3xMyc fusion (Supplementary Fig. 5a, b). Pull-down of C18orf8-3xMyc precipitated Ccz1 and Mon1B (Fig. 5c), and conversely, immune precipitation of endogenous Mon1B (Fig. 5c) or Ccz1 (Supplementary Fig. 5c) revealed the endogenous 3xMyc-tagged C18orf8. C18orf8 is composed of an N-terminal WD40 and C-terminal α-helical domain. To identify the MC1-interaction site in C18orf8, we expressed mScarlet-tagged single domains (Supplementary Fig. 5d). Immune precipitation revealed an interaction of Ccz1 and Mon1B with the C-terminal (AA 354-657), but not N-terminal (AA 1-362) domain of C18orf8 (Supplementary Fig. 5e).

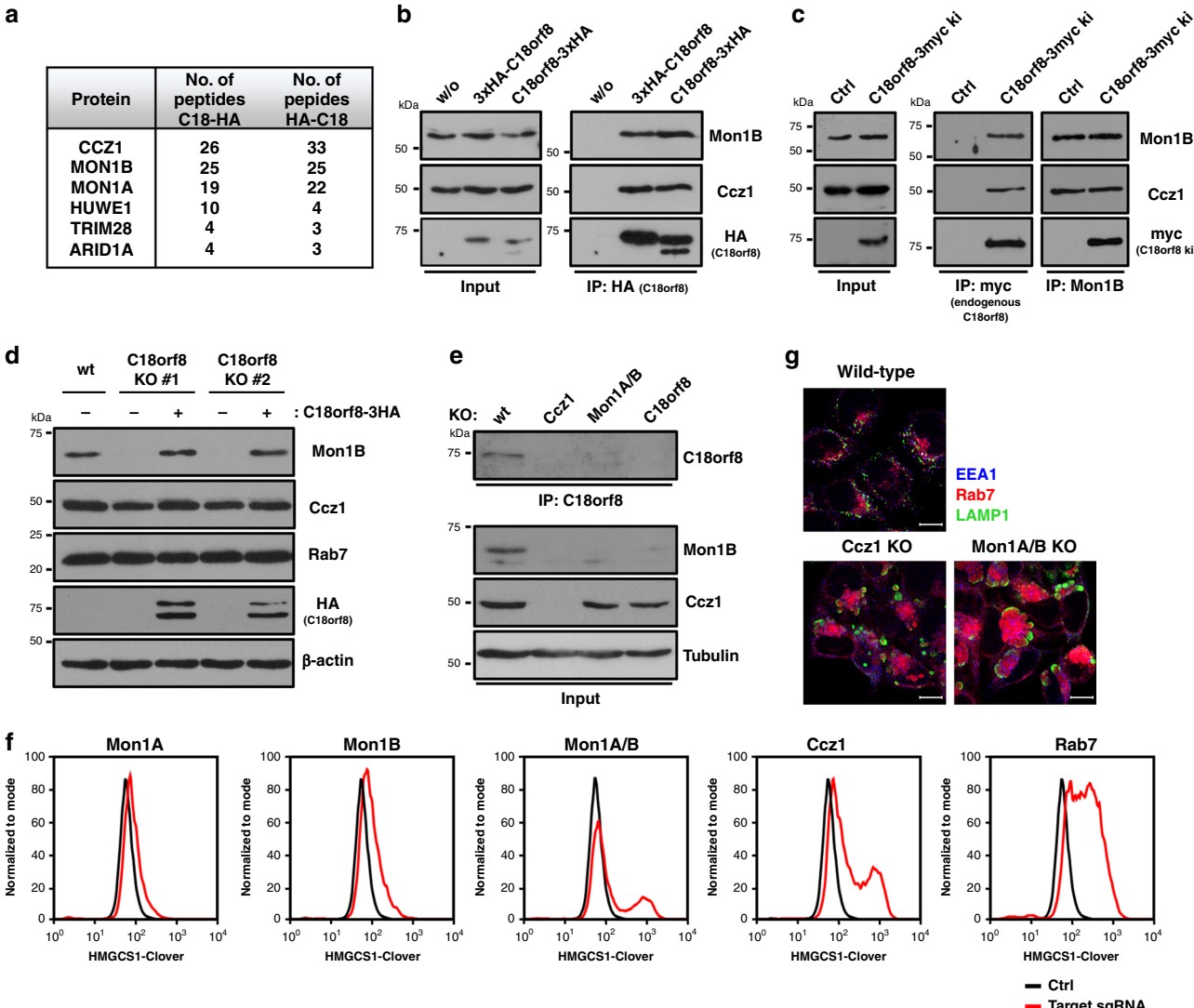

**Fig. 5 C18orf8 forms an integral component of the Mon1-Ccz1 (MC1) complex, essential for complex stability and function. a–c** C18orf8 interacts with the Mon1-Ccz1 complex. Immune precipitations of exogenous 3xHA-C18orf8 (N-term) or C18orf8-3xHA (C-term) were analysed by mass spectrometry (**a**) or Western blotting using Mon1B-, Ccz1- and HA-specific antibodies (**b**). Proteins detected by >3 peptides in both C18orf8 samples and absent from control, are indicated in **a**. Full MS results are available in Supplementary Data 2 and via ProteomeXchange with identifier PXD021444. **c** Immune precipitation of an endogenous C18orf8-3xMyc fusion or Mon1B reveals a reciprocal interaction between endogenous C18orf8, Ccz1 and Mon1B (see also Supplementary Fig. 5c). **d**, **e** C18orf8, Mon1B and Ccz1 show reciprocal stabilisation. Immunoblot analysis of lysates from (**d**) wild-type, *C18orf8*-deficient and *C18orf8*-deficient cells complemented with C18orf8-3xHA (3 independent experimental replicates); or (**e**) wild-type, *C18orf8*-, *Ccz1*- and *Mon1A/B*-deficient cells. For endogenous C18orf8 detection, C18orf8 was immune precipitated prior to immunoblotting (**e**). **f**, **g** *Ccz1*- and *Mon1A/B*-deficient cells show cholesterol deficiency and disruption of LE/Ly morphology. **f** HMGCS1-Clover CAS9 cells (black lines) were transfected with sgRNAs against *Mon1A*, *Mon1B*, *Mon1A* and *Mon1B*, *Ccz1* or *Rab7* (red lines) or control (black lines), grown for 14 days, treated overnight with mevastatin and analysed for HMGCS1-Clover expression. **g** Wild-type, *Ccz1*- and *Mon1A/B*-deficient cells were stained intracellularly for EEA1 (blue), Rab7 (red) and LAMP1 (green). Representative images are shown from five fields per condition. Scale bars = 10 μm.

The robust interaction of C18orf8 with all three MC1 components suggests C18orf8 is either a regulator or integral component of the mammalian MC1 complex. Interestingly, while *C18orf8*-deficient cells expressed normal levels of Ccz1, they showed a marked reduction in Mon1B expression (Fig. 5d), which was restored by re-expression of full-length C18orf8 (Fig. 5d), its C-terminal MC1-interaction domain (Supplementary Fig. 5e) or proteasome inhibition (Supplementary Fig. 5f). Mon1B expression was also decreased in *Ccz1*-deficient cells, while conversely endogenous C18orf8 expression was reduced in *Ccz1* and *Mon1A/B* double-deficient cells (Fig. 5e). Mon1B stability is therefore dependent on C18orf8 and Ccz1 and in the absence of either, Mon1B is proteasomally degraded. Similarly,

C18orf8 stability depends on Mon1A/B and Ccz1, whereas Ccz1 remains relatively stable in the absence of both other components (Fig. 5e). We conclude that mammalian Mon1, Ccz1 and C18orf8 form a stable trimeric complex, the MCC complex, of which C18orf8 is a core component.

*C18orf8*, *Ccz1* and *Rab7* are all prominent hits in our screens (Fig. 1g, Supplementary Fig. 2b). To determine whether the function of C18orf8 overlaps with mammalian MC1, we knocked-out *Ccz1*, *Mon1A*, *Mon1B* and *Rab7* in our *HMGCS1-Clover* reporter line. Similar to *C18orf8* depletion, loss of *Ccz1* or a combination of *Mon1A* and *Mon1B*—but neither alone—increased HMGCS1-Clover expression, as did knockout of the *Rab7* GTPase (Fig. 5f). This phenotype was restored by

complementation of knockouts with their respective wild-type proteins (Supplementary Fig. 5h), with *Mon1A/B*-double-deficient cells rescued by Mon1A or Mon1B alone. Like *C18orf8*-deficient cells, cells deficient for *Ccz1*- or *Mon1A/B* contained abnormal, enlarged Rab7+ (red) and LAMP1+ (green) LE/Ly compartments (Fig. 5g). This phenotypic similarity implies C18orf8 is required for both MCC stability and functionality. Importantly, although the C18orf8 C-terminal α-helical domain is necessary and sufficient for MCC binding and Mon1B stabilization (Supplementary Fig. 5e), it is insufficient for full complementation of the C18orf8-deficiency phenotype, which required full-length C18orf8 (Supplementary Fig. 5g). C18orf8 is therefore not simply a stabilizing factor but an active integral component, or subunit, of the mammalian MCC complex.

**The Mon1-Ccz1-C18orf8 complex is responsible for mammalian Rab7 activation.** Yeast MC1 acts as an activating GEF for the yeast Rab7 homologue Ypt7[28]. To assess whether the mammalian MCC complex has a similar function, we expressed HA-tagged wild-type, dominant-negative (T22N) and constitutively active (Q67L) mutants of Rab7 in C18orf8-3xMyc knock-in cells and probed their interaction with endogenous MCC components. Consistent with a role in Rab7 activation, C18orf8-3xMyc, Ccz1 and Mon1B preferentially bound the inactive Rab7-T22N, but not a constitutively active Rab7-Q67L mutant (Fig. 6a). This interaction is specific to Rab7 and does not occur with other dominant negative GTPases, including Rab5, Rab9, Rab11 and Arl8B (Supplementary Fig. 6a). Rab7 activation promotes binding and endosomal recruitment of its effectors RILP and ORP1L[17–20] and effector binding can be used to assess the activation status of Rab GTPases. In wild-type cells, an interaction between FLAG-tagged RILP and endogenous Rab7 was readily detected, indicating the presence of an active GTP-bound Rab7, whereas this interaction was completely lost in *C18orf8*-, *Ccz1*- and *Mon1A/B*-deficient cell lines (Fig. 6b). *MCC*-deficient cells are thus unable to activate Rab7, suggesting MCC acts as a mammalian Rab7 GEF. Indeed, HA-tagged RILP (green) was efficiently recruited to LAMP1+LE/Ly (magenta) in wild-type cells, but not in *C18orf8*-deficient cells, where RILP redistributed to the cytoplasm (Fig. 6c). LE recruitment of the Rab7 effector ORP1L was also strongly reduced (Supplementary Fig. 7a) and consistent with RILP's function in endosome mobilisation, LE motility was sharply decreased in *C18orf8*-deficient cells (Supplementary Fig. 7b). C18orf8, Ccz1 and Mon1A/B are therefore essential for Rab7 activation and functional recruitment of the Rab7 effectors RILP and ORP1L in mammalian cells.

**C18orf8 function can be by-passed by a constitutively active Rab7 or depletion of Rab7GAPs.** The activation of Rab GTPases by RabGEFs is counteracted by RabGAPs that enhance intrinsic GTPase activity and revert Rab GTPases back to an inactive state. Our data suggest that LE defects in *C18orf8*-deficient cells are secondary to defective Rab7 activation. We, therefore, asked whether LE function could be rescued by concomitantly blocking one or more Rab7GAPs; or by the use of a constitutively active Rab7-Q67L mutant that lacks intrinsic GTPase activity. Knockdown of the Rab7GAPs TBC1D5 and TBC1D15 (Fig. 6d, red line), but not either alone (green and blue lines), rescued HMGCS1-Clover levels in *C18orf8*-deficient cells back to wild-type (black line) and a similar rescue was obtained by expression of the constitutively active Rab7-Q67L (Fig. 6e, red line), but not the inactive Rab7-T22N (green line). Rab7-Q67L also restored the swollen LE/Ly phenotype (Fig. 6f, green) and DiI-LDL degradation (red) in *C18orf8*-deficient cells. In contrast, the T22N mutant augmented both phenotypes. Restoration of C18orf8 function was

specific to Rab7 and did not occur with constitutively active mutants of other endosomal GTPases, including Rab5, Rab9, Rab11 and Arl8B (Supplementary Fig. 6b). The trimeric Mon1-Ccz1-C18orf8 (MCC) complex therefore acts as an activating GEF for mammalian Rab7 and its role in late endosomal trafficking and LDL-cholesterol uptake can be specifically by-passed by a constitutively active Rab7 (Q67L) or by Rab7GAP depletion.

**MCC-deficient cells accumulate free cholesterol in their lysosomal compartment.** How does a defect in Rab7 activation result in cellular cholesterol deficiency? *C18orf8*-deficient cells showed a severe defect in LDL trafficking (Figs. 3e and 4e). We assumed this defect would impair cholesterol ester (CE) hydrolysis, decrease cholesterol release and, ultimately, cause cellular cholesterol deficiency. To ascertain the fate of LDL-derived cholesterol, we stained our cells with Filipin III, a bacterial compound that specifically binds free cholesterol and differentiates it from CEs in LDL[45]. Remarkably, *C18orf8*-deficient cells did not show a decrease in Filipin staining, but rather accumulated free cholesterol (green) in their swollen Rab7+ (blue) and LAMP1+ (red) LE/Ly compartment (Fig. 7a, b). Cholesterol accumulation resolved upon complementation with C18orf8-3xHA (Fig. 7a) and was also observed in *Ccz1*- and *Mon1A/B*-deficient cells (Fig. 7c). At the ultra-structural level cholesterol accumulation was confirmed by Theonellamides (TNM) staining[46,47] which enriched within the enlarged MVBs of *C18orf8*-deficient cells (Fig. 7d). The accumulation of free cholesterol in LE/Ly of *MCC*-deficient cells suggests that defective lysosomal cholesterol export, as opposed to LDL trafficking, is primarily responsible for the cellular cholesterol deficiency in *MCC*-deficient cells. Indeed the *MCC*-deficiency phenotype is strikingly similar to that observed in cells defective for the NPC1 cholesterol transporter (Fig. 7e).

**Active Rab7 interacts with the NPC1 cholesterol transporter.** The NPC1 transporter is critical for lysosomal export of LDL-derived cholesterol and cellular cholesterol homeostasis (Fig. 1g and Supplementary Figs. 2b and 3) and *NPC1* mutations are the primary cause of the Niemann Pick type C lysosomal storage disease. *C18orf8*-deficient cells show normal to elevated NPC1 expression (Supplementary Fig. 8a), with increased NPC1 expression in *Ccz1*- and *Mon1A/B*-deficient cells (Supplementary Fig. 8b). In wild-type cells, NPC1 resides predominantly in the LAMP1+ Ly compartment (Supplementary Fig. 8c), where it co-localises with NPC2 (Supplementary Fig. 8d) and this localisation was unaltered in *C18orf8*-, *Ccz1*- and *Mon1A/B*-deficient cells (Supplementary Fig. 8c, d). Since NPC1 expression and localisation appeared normal, we used mass spectrometry to identify NPC1 interactions partners. Remarkably, among the most abundant interaction partners identified in NPC1 immune precipitations is Rab7 itself (Fig. 8a and Supplementary Data 2, PXD021444). This interaction was confirmed in Rab7:NPC1 immune precipitations, in which NPC1 preferentially bound to the HA-tagged constitutively active Rab7-Q67L mutant, but not the inactive Rab7-T22N (Fig. 8b), as could be expected for a functionally important Rab7 interaction. Endogenous Rab7 equally interacted with endogenous NPC1 (Fig. 8c) and conversely endogenous NPC1 with endogenous Rab7 (Fig. 8d). Consistent with an activation-dependent event, the Rab7-NPC1 interaction seen in wild-type cells, was lost in *C18orf8*-, *Ccz1*- or *Mon1A/B*-cells that are defective in Rab7 activation (Fig. 8e). The Rab7-NPC1 interaction is maintained by an inactive NPC1-P692S mutant and is therefore independent of NPC1's cholesterol export function (Fig. 8f). Indeed Rab7 binding to NPC1 remained largely unaffected by treatment of cells with LPDS, which

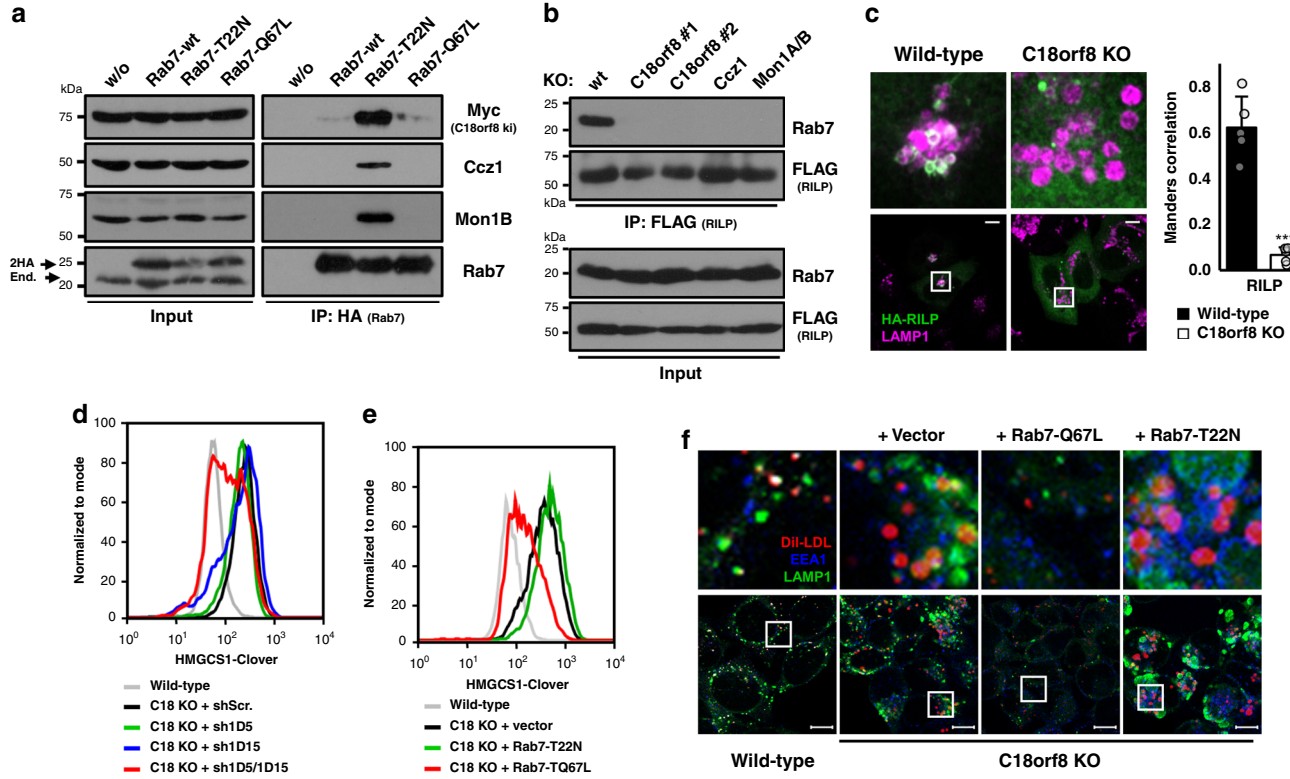

**Fig. 6 The trimeric Ccz1-Mon1-C18orf8 (MCC) complex activates mammalian Rab7. a** The MCC complex binds an inactive Rab7 (T22N). Immune precipitations of 2xHA-tagged wild-type, T22N or Q67L Rab7 from *C18orf8-3xMyc* knock-in cells, were analysed by immunoblot using Myc-, Ccz1- and Mon1B-specific antibodies (2 independent experimental replicates, see also Supplementary Fig. 6a). **b, c** *C18orf8-*, *Ccz1-* and *Mon1A/B*-deficient cells lack activation-dependent recruitment of Rab7 effectors. **b** Immune precipitations of 3xFLAG-RILP from wild-type, *C18orf8-*, *Ccz1-* or *Mon1A/B*-deficient cells were analysed by immunoblotting for endogenous Rab7 (two independent experimental replicates). **c** Wild-type and *C18orf8*-deficient cells were transfected with HA-RILP or ORP1L (Supplementary Fig. 7a) and stained intracellularly for HA (green) and LAMP1 (magenta). Mander's correlation was determined for $n = 5$ fields per condition from two independent experiments. Error bars reflect standard deviation (two-sided unpaired *t*-test, $p = 2.1 \times 10^{-5}$, ***$p < 0.001$). **d–f** Cholesterol and trafficking defects in *C18orf8*-deficient cells can be rescued by knockdown of Rab7GAPs or expression of a constitutively active Rab7 (Q67L). **d** HMGCS1-Clover expression was determined at day 8 after transduction of *C18orf8*-deficient cells with shRNAs against TBC1D5 (green), TBC1D15 (blue) or both (red). **e-f** *C18orf8*-deficient cells were transduced with 2xHA-tagged Rab7-T22N (green), -Q67L (red) or an empty vector (black) and either (**e**) analysed at day 10 by flow cytometry for HMGCS1-Clover expression (two independent experimental replicates, see also Supplementary Fig 6b), or **f** pulse-labelled with Dil-LDL (red), incubated for 3 h and stained intracellularly for LAMP1 (green) and EEA1 (blue). Representative images are shown from 5 fields per condition. Scale bars = 10 μm.

decreases, and the NPC1 inhibitor U18666A, which increases lysosomal cholesterol levels (Fig. 8g).

**Rab7 activation by the MCC GEF drives NPC1-dependent cholesterol export.** Loss of the Rab7-NPC1 interaction in *MCC*-deficient cells correlates with lysosomal cholesterol accumulation, suggesting Rab7 activation is essential for NPC1-dependent lysosomal cholesterol export. To test this hypothesis directly and avoid potential caveats created by LDL trafficking defects, we set up a lysosomal cholesterol export assay analogous to pulse-chase analysis (Fig. 9a). Cholesterol export was initially blocked using the reversible NPC1 inhibitor U18666A[48] to induce the accumulation of free cholesterol (green) in CD63+ LE/Ly (magenta) (Fig. 9b; pulse). The inhibitor was then washed out in LDL-free medium (LPDS) which allowed lysosomal cholesterol efflux in the absence of further LDL-cholesterol uptake (chase). Cholesterol release from LE/Ly was monitored by Filipin staining (Fig. 9c). During a 24 h chase period, cholesterol (green) was completely exported from CD63+ LE/Ly (magenta) in wild-type cells, whereas export was abolished in cells deficient for the NPC1 transporter (Fig. 9c, d). Similar to NPC1 deficiency, lysosomal cholesterol export was blocked in *C18orf8-*, *Ccz1-* and *Mon1A/B*-deficient cells in which Filipin staining remained entirely

co-localised with CD63 during the 24 h chase (Fig. 9c, d). To confirm that the cholesterol export defect in MCC-deficient cells depends on defective Rab7 activation, *C18orf8*-deficient cells were complemented with different Rab7 mutants. While the cholesterol accumulation resolved upon expression of the constitutively active Rab7-Q67L, no resolution was seen with either the wild-type or inactive Rab7-T22N (Fig. 9e), confirming Rab7 dependency of the cholesterol export defect.

NPC1 function is compromised in type C Niemann Pick (NPC) disease in which lysosomal cholesterol accumulates in multiple organs. The common NPC1[I1061T] mutation destabilizes the NPC1 transporter, resulting in its ER retention, and subsequent degradation[49]. However, any transporter reaching its correct lysosomal location remains functional[50]. We asked whether increased Rab7 activity might improve NPC1 function and promote clearance of lysosomal cholesterol in mutant NPC1 fibroblasts. Lentiviral overexpression of a GFP-tagged wild-type Rab7 indeed strongly reduced cholesterol accumulation in NPC1[I1061T/I1061T] patient fibroblasts (Fig. 9f), suggesting Rab7 activity is limiting for lysosomal cholesterol export in NPC disease.

Taken together our results show that Rab7 and its trimeric Mon1-Ccz1-C18orf8 (MCC) GEF control cellular cholesterol homeostasis through coordinated regulation of LDL trafficking

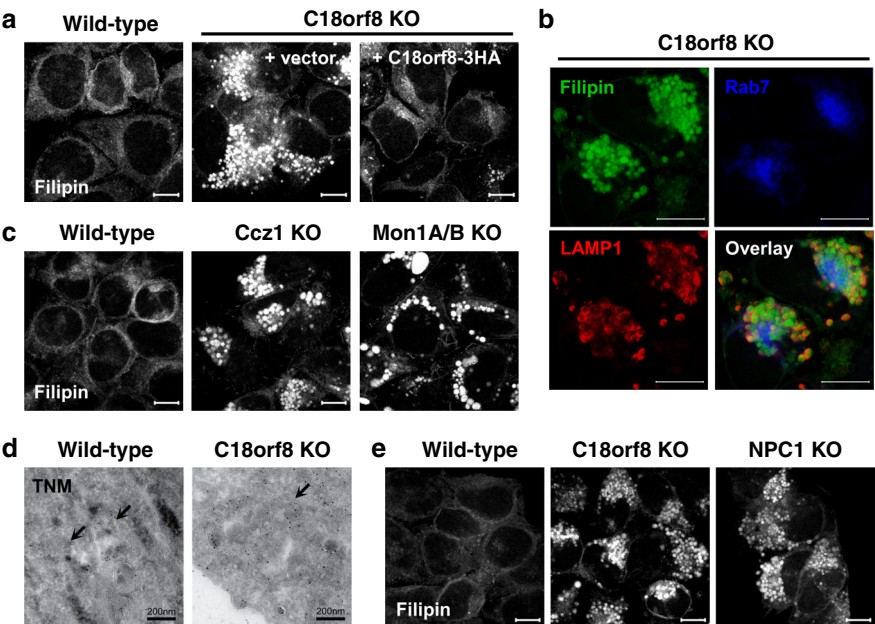

**Fig. 7 *C18orf8-*, *Ccz1-* and *Mon1A/B-* (MCC)-deficient cells accumulate free cholesterol in a swollen lysosomal compartment. a** Filipin staining of wild-type, *C18orf8*-deficient and complemented *C18orf8*-deficient cells; or **c** wild-type, *Ccz1-* and *Mon1A/B*-deficient cells. **b** Filipin (green) co-staining with the LE/Ly markers Rab7 (blue) and LAMP1 (red) in *C18orf8*-deficient cells. **d** Theonellamides (TNM) immuno-gold labelling of wild-type and *C18orf8*-deficient cells, visualised by EM. Multivesicular bodies (MVBs) are indicated by arrows. Note MVBs are markedly enlarged in *C18orf8*-deficient cells. **e** Filipin staining of wild-type, *C18orf8-* and *NPC1*-deficient cells. All confocal microscopy images are representative of five fields per condition and two independent experiments. Scale bars for confocal microscopy = 10 μm, scale bars for EM = 200 nm.

and NPC1-dependent lysosomal cholesterol export, making it a potential therapeutic target in Niemann Pick disease.

## Discussion

The endocytic uptake of LDL-cholesterol is a major source of cellular cholesterol. Using an endogenous cholesterol sensor, HMGCS1-Clover, in a genome-wide and secondary sub-genomic CRISPR library screen we identified >100 genes involved in the cellular LDL-cholesterol uptake pathway. Of these, we characterised C18orf8 as a novel core component of the Mon1-Ccz1 (MC1) GEF for mammalian Rab7. *C18orf8*-deficient cells are defective for Rab7 activation, resulting in impaired LDL trafficking and cellular cholesterol depletion. The unexpected accumulation of free cholesterol in the swollen LE/Ly compartment of *C18orf8*-deficient cells, suggested an additional critical role for Rab7 in cholesterol export out of lysosomes. We show that active Rab7 binds the NPC1 cholesterol transporter and licenses lysosomal cholesterol export. This pathway is defective in *C18orf8-*, *Ccz1-* and *Mon1A/B*-deficient cells and restored by a constitutively active Rab7 (Q67L). The trimeric M̲on1-C̲cz1-C̲18orf8 (MCC) complex thus plays a central role in cellular LDL-cholesterol uptake, coordinating Rab7 activation, LDL trafficking and NPC1-dependent lysosomal cholesterol export.

Genome-wide CRISPR screens provide a powerful tool to interrogate intracellular pathways. To avoid artefacts associated with exogenous reporters, we generated an endogenous SREBP2 knock-in reporter, HMGCS1-Clover, and screened for genes regulating cellular cholesterol homeostasis. Hits from our primary genome-wide screen were validated in a secondary targeted screen using a newly generated sub-genomic sgRNA library containing the top 1000 ranking genes from the primary screen. This approach confirms the reproducibility of our screening results with 52 out of 65 secondary screening hits present in our primary screen, while 41 hits were unique to the primary

genome-wide screen and 13 hits unique to the secondary targeted screen (Fig. 2b). As no single genetic screening approach is sufficient to give a complete picture of a complex cellular pathway, our two screens are complementary as well as overlapping, with multi-subunit complexes identified across screens. For example, while our genome-wide screen hits the COG subunits COG3/4/7 and V-ATPase subunits ATP6AP1/AP2/V0D1/V0E1/V1A/V1B2/V1G1/V1H, our targeted screen identifies additional subunits (COG5/8, ATP5V1D/V1F) and the V-ATPase assembly factors CCDC115, TMEM199 and VMA21. Conversely, our targeted screen only identifies the AP2 μ1 subunit (AP2M1), whereas our genome-wide screen also identifies the AP2 σ1 subunit (AP2S1), the assembly factor AAGAB and accessory factor FCHO2.

The predominant pathway identified by both of our screens is LDL-cholesterol import, as HeLa cells, like many tissue-culture cells, rely predominantly on cholesterol import rather than de novo biosynthesis. The screens identified multiple stages in the LDLR-LDL trafficking pathway (Fig. 2c), including (i) LDLR folding and trafficking to the cell surface; (ii) endocytosis of the LDLR-LDL complex; (iii) LDL trafficking to the LE/Ly compartment; and (iv) NPC1 dependent cholesterol export from LE/Ly. Functional complexes identified include COG, Exocyst, AP2, V-ATPase, PI3K, PIKfyve, CORVET, HOPS, FERARI and GARP; poorly characterised genes include the chaperone Clusterin (also known as Apolipoprotein J) and the metalloprotease ADAM28. Our genome-wide and targeted CRISPR screens thus provide an extensive overlapping dataset of 106 genes required for LDL-cholesterol import, and identify novel candidate genes in this pathway.

Spatial and temporal control of Rab7 activation is critical to coordinate substrate trafficking and degradation in LE/Ly[16]. In yeast the dimeric Mon1-Ccz1 (MC1) GEF activates the Rab7 homologue Ypt7[27,28] and a similar function has been suggested for mammalian MC1[35,36]. We find that a stable trimeric complex of Mon1A/B, Ccz1 and C18orf8 (MCC) is required for Rab7

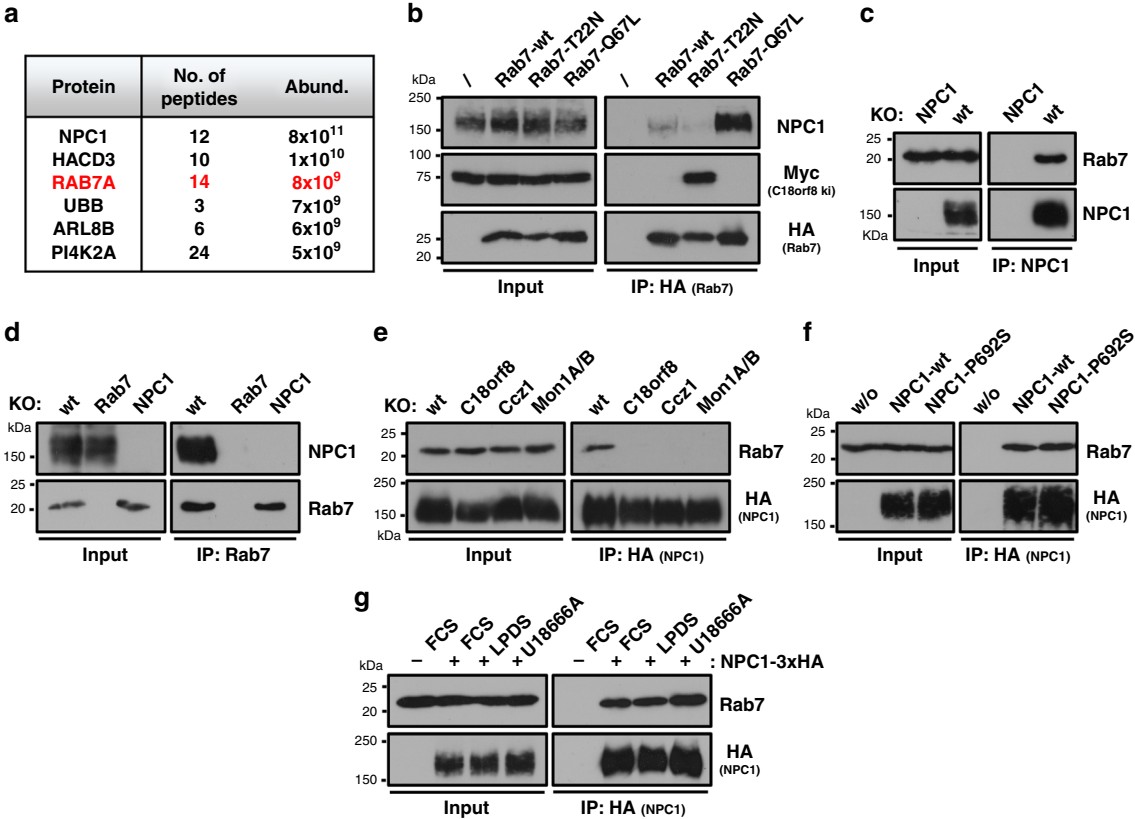

**Fig. 8 Rab7 interacts with the lysosomal cholesterol transporter NPC1 in an activation-dependent manner. a** Immune-precipitation of HA-tagged NPC1 and detection of NPC1-interacting proteins using mass spectrometry. Most abundant interaction partners detected with >2 peptides are indicated (full dataset available in Supplementary Data 2 and via ProteomeXchange with identifier PXD021444). **b** Immune precipitations of HA-tagged wild-type, dominant-negative (T22N) or constitutively active Rab7 (Q67L) show an activation-dependent interaction between Rab7 and endogenous NPC1 (two independent experimental replicates). **c, d** Reciprocal immune-precipitation of endogenous NPC1 (**c**) or Rab7 (**d**) shows a strong interaction between both proteins. **e** The Rab7-NPC1 interaction is lost in *MCC*-deficient cells that lack Rab7 activation. Wild-type, *C18orf8-*, *Ccz1-* and *Mon1A/B*-deficient cells were stably transduced with the inactive NPC1-P692S-HA. HA-tagged NPC1 was immune precipitated and immune blotted for endogenous Rab7. The inactive NPC1-P692S was used to prevent altering lysosomal cholesterol content by NPC1 overexpression. (two independent experimental replicates) **f, g** The Rab7-NPC1 interaction is independent of NPC1 activity or lysosomal cholesterol levels. **f** *NPC1*-deficient cells were complemented with HA-tagged wild-type or inactive P692S-mutant NPC1 and NPC1-HA immune-precipitations were analysed by immune blotting for endogenous Rab7. **g** Wild-type NPC1-HA complemented cells were treated with LPDS to decrease, or U18666A to increase lysosomal cholesterol levels and the NPC1-Rab7 interaction was probed using immune precipitation (two independent experimental replicates).

activation in mammalian cells, as defined by effector recruitment. *C18orf8*, *Mon1* and *Ccz1* are conserved from animals to plants, but a *C18orf8* orthologue is missing from laboratory yeast and most other fungi, either due to divergence of the endocytic machinery or because C18orf8 function is incorporated into other components.

Rab7 is a key regulator of LE homeostasis and *C18orf8-*, *Ccz1-* and *Mon1A/B*-deficient cells show a broad range of LE defects including LE morphology, endosomal substrate trafficking, lysosomal degradation and cholesterol export. Consistent with a previous Rab7 depletion study[51], *C18orf8*-deficient cells accumulate enlarged MVBs; and consistent with current Rab5-to-Rab7 conversion models[32] we observe a delay in early-to-late endosomal trafficking. Whereas LDL trafficking was previously suggested to involve only Mon1B[52], we find that only the combined depletion of Mon1A/B causes cholesterol deficiency, suggesting redundancy of Mon1 homologues.

Besides LE trafficking, Rab7 is an important factor in autophagy[53]. During preparation of this manuscript, a role for C18orf8 was reported as a regulator of autophagic flux[54]. Like us, this group reports a robust interaction between C18orf8 and mammalian MC1. However, without direct evidence on Rab7

activation, C18orf8 is postulated to be a MC1 regulator. We find C18orf8 is not a regulator, but a core subunit of the trimeric mammalian MCC complex, which acts as a Rab7 GEF. Stability of the three core components is interlinked, with Mon1B stability dependent on C18orf8 and Ccz1, and C18orf8 stability dependent on Ccz1 and Mon1B. Ccz1 appears relatively stable in the absence of both other components and may therefore form a scaffold for Mon1A/B and C18orf8 assembly. The intricate relationship between C18orf8 and Mon1A/B is intriguing as Phyre2 prediction suggests C18orf8 adopts a clathrin-like fold, while Mon1A/B shows structural similarity to the AP2μ subunit[55]. Whether this structural similarity has functional consequences, remains to be established.

The structural interdependence of the MCC subunits complicates in vivo studies to assess the direct role of C18orf8 in mammalian MCC. Yeast Ccz1p and Mon1p are necessary and sufficient for Ypt7 binding and GDP-to-GTP exchange, with most of the MC1-Ypt7 binding interface mediated by Mon1p[28,34]. Residues critical for Ypt7 binding are conserved in mammalian MC1 and a dimeric complex of mammalian Mon1A and Ccz1 was shown to have limited in vitro Rab7 GEF activity[36]. While this complex might have retained residual C18orf8 due to

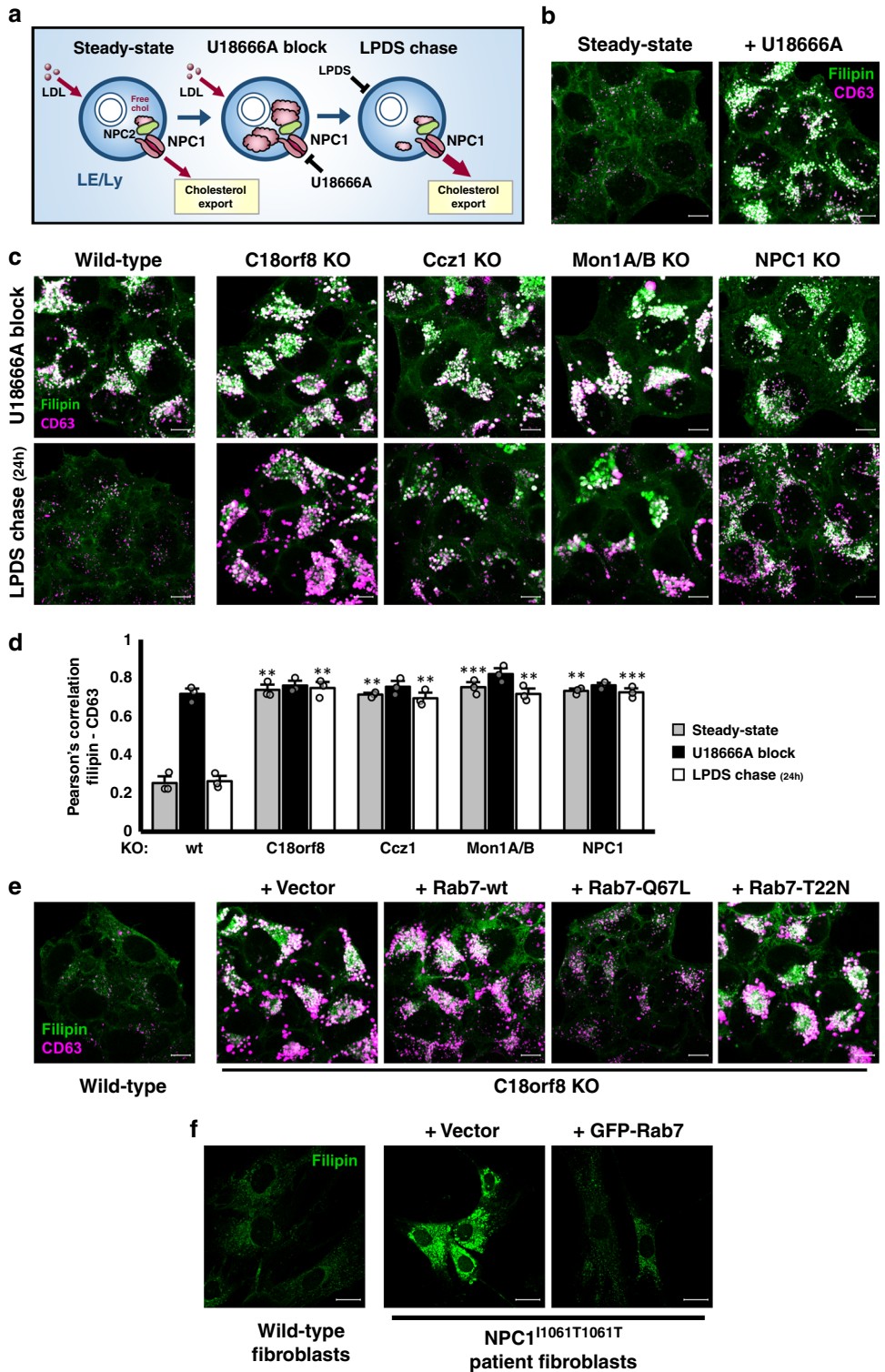

its isolation from mammalian cells, future studies will address whether C18orf8 contributes towards in vitro MC1 GEF activity.

In vivo, C18orf8 function goes beyond being a scaffold for MCC stabilisation. The C18orf8 C-terminal α-helical domain is sufficient to bind MC1 and stabilize Mon1B, but insufficient to fully restore MCC function (Supplementary Fig. 5e–g). The C18orf8 N-terminus thus has an independent function and may bind other trafficking components and/or membrane lipids. The timing of Rab7 activation is strictly regulated to coordinate Rab5-to-Rab7 conversion and likely requires control of MCC by

upstream components[32]. Membrane association allosterically activates yeast MC1 and increases its GEF activity ~1600-fold[29]. Like yeast Mon1p and Ccz1p, mammalian Mon1A binds PtdIns3P and PtdSer[27], while positively charged residues in the N-terminal β-propeller of C18orf8 are well positioned to bind negatively charged phospholipids. In analogy to AP2[56], PtdIns binding might open the MCC complex towards Rab7 binding, thus allosterically activating MCC upon membrane association.

Cholesterol is hydrolysed from CEs in a LE/Ly compartment, and NPC1 is essential for its subsequent export to the ER,

**Fig. 9 Rab7 activation by the MCC GEF controls NPC1-dependent lysosomal cholesterol export. a–d** Lysosomal cholesterol export is abolished in *NPC1-*, *C18orf8-*, *Ccz1-* and *Mon1A/B*-deficient cells. **a** Schematic depiction of the cholesterol export assay. Similar to a pulse-chase analysis, LDL-derived free cholesterol is allowed to accumulate in a lysosomal compartment using the NPC1 inhibitor U18666A (pulse) and released during the LPDS chase. **b** Free cholesterol (Filipin, green) accumulates in a CD63+ LE/Ly compartment (magenta) of wild-type HeLa cells following 24 h incubation with U18666A. **c** Wild-type, *NPC1-*, *C18orf8-*, *Ccz1-* and *Mon1A/B*-deficient cells were treated for 24 h with U18666A (top panels), followed by a 24 h chase in the presence of LPDS and mevastatin (lower panels). Lysosomal cholesterol accumulation was visualised using Filipin (green) co-staining with the LE/Ly marker CD63 (magenta). **d** Co-localisation of Filipin with CD63 is plotted as Pearson correlation, calculated from $n = 3$ independent experiments with six fields per condition per experiment and >8 cells per field. Error bars reflect standard error of mean (two-side paired *t*-test, **$p < 0.01$, ***$p < 0.001$). **e** Cholesterol accumulation in *C18orf8*-deficient cells is abolished by overexpression of a hyperactive Rab7. *C18orf8*-deficient cells were transduced with a wild-type, dominant-negative (T22N) or constitutively active Rab7 (Q67L) or empty vector and co-stained with Filipin (green) and anti-CD63 (magenta). Representative images are shown from five fields per condition. **f** Rab7 overexpression rescues lysosomal cholesterol accumulation in NPC patient fibroblasts. NPC[I1061T/I1061T] primary patient fibroblasts were transduced with a GFP-tagged wild-type Rab7 or vector control and analysed at day 7 for cholesterol accumulation using Filipin staining. Fibroblasts from a healthy individual were used as a control. Representative images are shown from five fields per condition and two independent experiments (see also Supplementary Fig. 9). Scale bars = 10 μm.

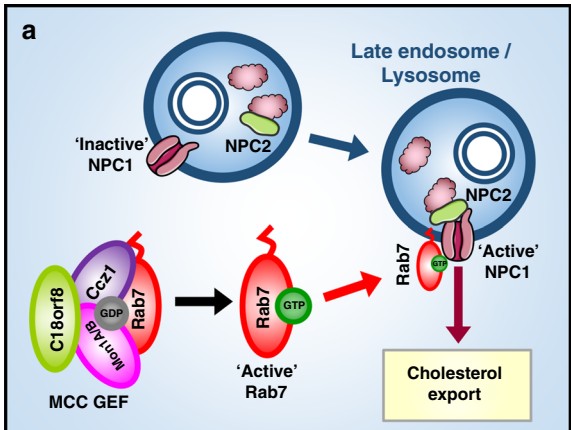 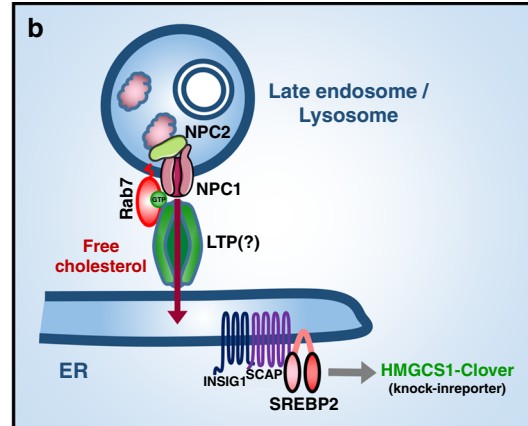

**Fig. 10 A trimeric Rab7 GEF controls NPC1-dependent lysosomal cholesterol export.** Model for MCC and Rab7 function in lysosomal cholesterol export. The trimeric Mon1-Ccz1-C18orf8 (MCC) GEF activates mammalian Rab7, which binds the NPC1 cholesterol transporter and either **a**) directly activates NPC1's cholesterol export function; or **b**) assembles a down-stream membrane contact site (MCS) at which a yet-uncharacterised lipid transfer protein (LTP) mediates cholesterol transfer to the ER and/or plasma membrane. A combined Rab7 function in NPC1 activation and MCS formation would assure lysosomal cholesterol export only proceeds once a down-stream lipid transfer module is assembled.

mitochondria and plasma membrane. Increased expression of our SREBP2 reporter indicates *C18orf8-*, *Ccz1-* and *Mon1A/B*-deficient cells are starved of ER cholesterol, yet paradoxically they accumulate free cholesterol in a swollen LE/Ly compartment. This suggested a novel role for Rab7 in lysosomal cholesterol export. We find an active Rab7 binds the lysosomal cholesterol transporter NPC1 and licenses its export of LDL-derived cholesterol from LE/Ly. The inability of MCC-deficient cells to activate Rab7, results in a Niemann-Pick-like phenotype, even though NPC1/NPC2 expression and localisation are intact. Rab7, therefore, functions as a novel regulator of NPC1 and its export of lysosomal cholesterol.

The mechanism by which Rab7 controls NPC1-dependent lysosomal cholesterol export remains at present unclear. There is however at least three ways in which Rab7 might regulate this key process: (i) Rab7 may directly activate the NPC1 transporter, thus licensing lysosomal cholesterol export (Fig. 10a); (ii) Rab7 might assemble a lipid transfer hub down-stream of NPC1 to allow inter-organelle cholesterol transfer at membrane contact sites (MCS) (Fig. 10b); or (iii) Rab7 may act by a combination of both.

The activity of many transporters is tightly regulated as exemplified by the lysosomal calcium channel TRPML1, which is activated at LE/Ly by PtdIns(3,5)P₂[57]. Even so, the NPC1 transporter is believed to be constitutively active and to date no NPC1 regulators have been identified. So why would Rab7 regulate NPC1 activity? Inappropriate cholesterol export by a nascent NPC1 could harm the composition and functionality of non-

lysosomal compartments. To prevent cholesterol export outside LE/Ly, the cholesterol hand-off between NPC2 and NPC1 relies on an acidic lysosomal pH[7]. Rab7's activation of NPC1-dependent cholesterol export could further restrict cholesterol transfer to the appropriate LE/Ly compartment (Fig. 10a). The Rab7-NPC1 interaction is independent of lysosomal cholesterol content. Even so, variations in Rab7 activity might tune lysosomal cholesterol export to cellular or organelle-specific conditions, such as MCS formation or organelle division. Future studies will aim to identify the cytoplasmic Rab7 binding site on NPC1 and its putative function in regulating NPC1 activity.

Besides NPC1 activation, Rab7 might control lysosomal cholesterol export through the formation of inter-organelle MCS and/or the recruitment of lipid transfer proteins (LTPs) down-stream of NPC1 (Fig. 10b). At least three Rab7 effectors - ORP1L, Protrudin (ZFYVE27) and PDZD8 – have been involved in ER-to-LE MCS formation[19,20,58,59] and Rab7 itself has been implicated in ER-to-mitochondria MCS formation[60]. MCS are of key importance for rapid inter-organelle lipid exchange, yet whether Rab7-mediated MCS are involved in cholesterol transfer remains controversial. The Rab7 effector ORP1L was reported to act as a LTP linking the ER and LE/Ly and transporting cholesterol towards the ER in a PtdIns(4,5)P₂ / PtdIns(3,4)P₂-dependent manner[12,61]. ORP1L depletion, however, causes only minor steady-state changes in lysosomal cholesterol, as does depletion of other LTPs and tethering proteins such as STARD3, ORP5 and SYT7[13,14,62]. By contrast, *C18orf8-*, *Ccz1-* and *Mon1A/B*-deficient

cells are entirely defective in lysosomal cholesterol export and fully phenocopy an NPC1-deficiency phenotype (Figs. 7d and 9c, d). This places Rab7 firmly at the centre of lysosomal cholesterol export and suggests the LTP(s) involved might be Rab7 effectors. Future studies will aim to identify the full set of (redundant) Rab7-effectors and LTPs involved in lysosomal cholesterol export.

LDL-cholesterol uptake is a highly efficient process, with each LDLR molecule estimated to endocytose one LDL particle containing ~1600 CE/cholesterol molecules every 10 min[3]. Down-stream LDL processing must be tightly coordinated to prevent a backlog in the endocytic pathway. CE hydrolysis likely commences *en route* to NPC1 + Ly to allow the large LDL particle to unpack; whereas the arrival and export of cholesterol in Ly must be linked to the formation and transport capacity of MCS. As a central regulator or LE homeostasis, Rab7 is well suited to coordinate these processes. Indeed, *MCC*-deficient cells show defects in LDL trafficking and lysosomal cholesterol export, indicating both are linked to the Rab7 nucleotide cycle. We propose that Rab7 and its MCC GEF are central regulators of cellular LDL-cholesterol uptake, coordinating LDL trafficking, NPC1-dependent lysosomal cholesterol export and lipid transfer at MCS. Besides cholesterol, the Rab7 lipid transfer hub might mediate lysosomal egress of other lipids, with VPS13A/C prominent Rab7 effectors[63,64].

Mutations in NPC1 and NPC2 cause the lethal Niemann Pick type C (NPC) lysosomal storage disease. Disease-associated mutations commonly destabilize NPC1, leading to its ER retention and proteasome- or lysosome-mediated degradation[49,65]. The most prevalent NPC1 mutation, I1061T, destabilizes NPC1, yet the mutant protein retains its cholesterol export function and the disease phenotype can be restored by overexpression of the mutant NPC1[50]. Confirming earlier observations[66], we show lentiviral overexpression of a GFP-tagged wild-type Rab7 reduces cholesterol accumulation in NPC1[I1061T/I1061T] patient-derived primary fibroblasts (Fig. 9f). Whereas this rescue was previously ascribed to enhanced vesicular trafficking, we suggest that Rab7 overexpression directly enhances NPC1-dependent lysosomal cholesterol export. This effect depends on an active Rab7 and is not observed with the inactive Rab7-T22N, whereas the hyperactive Rab7-Q67L shows an intermediate phenotype (Supplementary Fig. 9). A modest increase in Rab7 activity therefore increases lysosomal cholesterol clearance in NPC1-mutant cells while optimal clearance requires both Rab7 activation and inactivation in a time- and location-dependent manner.

In summary, we have identified a novel function for the Rab7 GTPase and its trimeric Mon1-Ccz1-C18orf8 (MCC) GEF in the coordination of late endosomal LDL trafficking and NPC1-dependent lysosomal cholesterol export. Future studies will aim to further characterise the mechanism behind Rab7 function in lysosomal cholesterol export and explore its therapeutic potential in Niemann Pick disease.

## Methods

**Materials**. Details of all plasmids, shRNA and sgRNA constructs, gene fragments, primers and antibodies used in this study are indicated in the Supplementary Methods. Primers were purchased from Sigma-Aldrich, gene fragments from Integrated DNA Technologies (IDT).

**Cell lines**. HeLa and 293T cells were maintained respectively in RPMI-1640 and IMDM (Sigma-Aldrich) supplemented with 10% foetal calf serum (FCS). For sterol depletion, cells were washed in PBS and incubated overnight in RPMI supplement with 5% lipoprotein-depleted serum (LPDS, Biosera), 10μM mevastatin (Sigma-Aldrich) and 50μM mevalonate (Sigma-Aldrich). NPC1[I1061t/I1061T] patient-derived primary fibroblasts (GM18453) and healthy controls (GM08399) were obtained from Coriell Institute for Medical Research (New Jersey, USA) and grown in MEM with Earl's salt (Sigma-Aldrich) supplemented with 15% FCS and non-essential amino acids (Gibco).

CRISPR knock-out lines were created by transient transfection of HeLa cells using TransIT HeLa Monster (Mirus). Transfected cells were selected on puromycin 24–72 h post-transfection and single-cell cloned >7 days post transfection. Knock-out clones were characterised by flow cytometry and Western blotting.

Stable protein overexpression was achieved using lentiviral transduction. Briefly, 293T cells were co-transfected in a 1:1 ratio with a lentiviral expression vector (pHRSIN/pHRSiren/pKLV) and the packaging vectors pMD.G and pCMVR8.91 using TransIT-293 (Mirus). The supernatant was harvested at 48 h post-transfection and transferred onto target cells. Cells were spun 60 min at 700 g to enhance infectivity and incubated with virus overnight. Transduced cells were selected for stable transgene expression with appropriate antibiotics from 48 h post-transduction.

**CRISPR knock-in of C-terminal tags**. HeLa cells were transiently transfected with sgRNA, CAS9 and pDonor vector using TransIT HeLa Monster (Mirus). To create a C18orf8-3xMyc knock-in a single donor vector (pDonor C18orf8-3xMyc Hygro, Supplementary Fig. 5a) was used, whereas for HMGCS1-Clover knock-in a combination of three donor plasmids (pDonor HMGCS1-Clover Puro, Hygro, Blast, Supplementary Fig. 1a) was used to select for multiple-allele integration. Transfected cells were transiently selected for sgRNA expression using puromycin at 24–72 h post-transfection, followed by selection of the knock-in cassette using hygromycin or a combination of puromycin, hygromycin and blasticidin at 7 days post-transfection. Selected cells were transiently transfected with Cre recombinase to remove resistance cassettes and single-cell cloned. Single-cell clones were characterised using flow cytometry and Western blotting.

**Cloning of the top 1000 sub-genomic sgRNA library**. The top 1000 sub-genomic sgRNA library was cloned as detailed previously[67,68]. sgRNA sequences for the top 1000 genes of the genome-wide CRISPR screen were retrieved from the Bassik genome-wide CRISPR library (10 sgRNAs/gene, Supplementary Data 1)[39]. Duplicate sgRNAs and sgRNAs with internal BsmBI sites were replaced with sgRNAs derived from the genome-wide Sabatini library[69] or designed de novo using the Broad Institute sgRNA designer (https://portals.broadinstitute.org/gpp/public/analysis-tools/sgrna-design). Targeting sgRNAs (total 9936) were appended with the overhangs 5′- AGGCACTTGCTCGTACGACGCGTCTCACACCG-sgRNA(17-20nt)- GTTTCGAGACGATGTGGGCCCGGCACCTTAA-3′ and 2500 non-targeting sgRNAs (derived from the genome-wide Bassik library) were added with the overhangs 5′- GTGTAACCCGTAGGGCACCTCGTCTCACACCG-sgRNA(17-20nt)-GTTTCGAGACGGTCGAGAGCAGTCCTTCGAC-3′. The pooled oligo library (total 12,346 sgRNAs, Supplementary Data 1) was synthesised by GenScript (Netherlands). Targeting and non-targeting oligos were amplified separately using Q5 hot-start polymerase (NEB) and primers indicated in the Supplementary Materials. PCR products were digested with BsmBI (Thermo-Fisher), ligated in a BsmBI-digest modified pKLV.U6 pGK Puro-2A-BFP vector and transformed at >30-fold coverage in ElectroMax Stbl4 competent cells. Plasmid DNA was prepped using a Qiagen Plasmid Maxi Kit.

**Genome-wide and targeted sub-genomic CRISPR screening**. HeLa HMGCS1-Clover cells were stably transduced with Cas9. CRISPR sgRNA library lentivirus was produced as indicated above and titrated on HeLa HMGCS1-Clover Cas9 cells. For genome-wide CRISPR screening, $1 \times 10^8$ HeLa HMGCS1-Clover Cas9 cells were transduced at 30% infectivity (>100-fold coverage) with the genome-wide Bassik library[39], puromycin selected, sorted at day 9 ($7.4 \times 10^7$ cells at 0.56%) and again at day 18 ($1.9 \times 10^7$ cells at 3.1%) for a HMGCS1-Clover[high] phenotype. Sorted cells were grown for another 5 days, harvested ($5 \times 10^6$ cells) and genomic DNA was extracted using a Gentra Puregene Core kit A (Qiagen).

For targeted sub-genomic CRISPR screening $2.4 \times 10^7$ HeLa HMGCS1-Clover CAS9 cells were transduced at 28% infectivity (>500-fold coverage) with the top 1000 sgRNA library, puromycin selected, sorted at day 8 ($5.2 \times 10^7$ cells at 1.1%) and again at day 12 ($1.5 \times 10^7$ cells at 6.7%) for a HMGCS1-Clover[high] phenotype and genomic DNA was extracted from sorted cells ($5 \times 10^5$) and an age-matched library sample ($1 \times 10^7$ cells) at day 12.

Integrated sgRNA was amplified via two rounds of PCR using primers as indicated in Supplementary Methods, starting with 400 μg and 100 μg of gDNA respectively for the library and sorted sample of the genome-wide CRISPR screen and 200 μg and 14 μg gDNA for the library and sorted sample of the sub-genomic CRISPR screen. PCR products were purified using AMPure XP beads (Agencourt), quantified on a DNA-1000 chip (Agilent) and sequenced on a Miniseq sequencer (Illumina) running MiniSeq Control Software v1.1.8 (Illumina). Reads were aligned to library sequences using Bowtie[70], allowing read alignment to a maximum of 2 sgRNA and 1 mismatch. sgRNA abundance in the sorted sample was compared against a library sample of similar age (genome-wide screen) or a screen-internal library sample (targeted screen) and sgRNA enrichment was computed using the MAGeCK algorithm under default settings[71]. Enrichment and de-enrichment of individual targeting and non-targeting sgRNAs is plotted in Supplementary Fig. 10. Hits with a MAGeCK score $<10^{-5}$ (genome-wide screen) or $<10^{-4}$ (targeted screen) were manually annotated into functional pathways (cholesterol import, endocytosis, endosomal trafficking, ER/Golgi trafficking, protein folding/

glycosylation, SREBP pathway) (Fig. 2c). Full datasets, including read counts, MAGeCK analysis and selected hits, are available in Supplementary Data 1.

**Immunoblotting and Immunoprecipitation.** For immunoblotting, cells were lysed for 30 min on ice in 1% IGEPAL (Sigma-Aldrich) or 1% Digitonin (Merck) in TBS pH 7.4 with Roche complete protease inhibitor or for 30 min at RT in 2% SDS in 50 mM Tris pH7.4 in the presence of Benzonase (Sigma-Aldrich). Post-nuclear supernatants were heated 15 min at 50 °C in SDS sample buffer, separated by SDS-PAGE and transferred to PVDF membranes (Millipore). Membranes were probed with the indicated antibodies and reactive bands visualised with ECL, Supersignal West Pico or West Dura (Thermo Scientific).

For immunoprecipitations, cells were lysed for 30 min on ice in 1% Digitonin, 1% IGEPAL or 0.5% IGEPAL in TBS pH 7.4 with Roche protease inhibitor. Samples were precleared with protein A/IgG-Sepharose and incubated with primary antibody and protein A/protein G-Sepharose or antibody conjugated agarose for at least 2 h at 4 °C. After five washes in 0.2% detergent, proteins were eluted in 2% SDS, 50 mM Tris pH7.4, separated by SDS-PAGE and immunoblotted as described. For RILP/NPC1/Rab7 immune precipitations, lysis buffers were supplemented with 10 mM MgCl₂ and 1 mM EDTA.

**Metabolic labelling and pulse-chase.** Cells were sterol-depleted overnight, starved for 30 min at 37°C in methionine-free, cysteine-free RPMI containing 5% dialysed LPDS, labelled with [35S]methionine/[35S]cysteine) (Amersham) for 10 min and then chased in RPMI with 10% FCS supplemented with 2 µg/ml 25-hydroxy-cholesterol and 20 µg/ml cholesterol. Samples taken at the indicated time-points were lysed in 1% Digitonin/TBS as above. Immunoprecipitations were performed as above, washed with 1% Tx-100/TBS, eluted and samples separated by SDS-PAGE and processed for autoradiography with a Cyclone scanner (Perkin-Elmer).

**Flow cytometry.** Cells were washed in PBS, detached by trypsinising for 5 min, pelleted and where necessary antibody stained for 30 min on ice. After washing in ice-cold PBS, cells were analysed on a FACS Calibur or FACS Fortessa (BD Biosciences) using BD CellQuant v5.2 and FACSDiva v8.0.1 software, respectively, and analysed in FlowJo v10.6.1(LCC).

**Immune fluorescence.** Cells were grown on 10 mm coverslips, fixed for 15 min in 4% formaldehyde (Polysciences), washed three times with PBS and incubated 10 min with 15 mM Glycine. Cells were permeabilised for 1 h with 0.05% saponin, 5% goat serum in PBS and stained for 1–2 h with primary antibody as indicated in 3% BSA, 0.05% saponin, PBS. Coverslips were washed three times 5 min in 0.1% BSA, 0.05% saponin, PBS and incubated 1 h with fluorochrome-conjugated secondary, washed 3 more times and embedded in Prolong Antifade Gold (Thermo Scientific). Stainings with Rab5 and Rab7 antibody (Cell Signalling) were performed overnight in 3% BSA, 0.3% Tx-100, PBS at 4 °C, followed by staining with additional primary antibody and secondary antibodies as above.

For Dil-LDL co-staining, cells were permeabilised for 40 s in 0.01% Digitonin (Merck) and antibody stained in 3% BSA without detergent. For Filipin staining, cells were fixed and incubated 1 h with 0.05 mg/ml Filipin III (Cayman) in 0.5% BSA. For Filipin co-staining, primary and secondary antibody stainings were performed in the presence of Filipin and without detergent. Fluorescent staining was recorded on a Zeiss LSM880 Confocal microscope using a 63× oil objective and analysed using Zen v2.3 software (Zeiss).

For ORP1L/RILP-LAMP1 co-localisation, cells were transiently transfected with HA-tagged RILP or ORP1L constructs using Effectine (Qiagen), harvested at 24 h post transfection and fixed in 3.7% formaldehyde in PBS for 15–20 min. After 10 min permeabilised with 0.1% TritonX-100 in PBS, coverslips were stained with HA and LAMP1 primary antibodies and donkey anti-rat CF568 and donkey anti-mouse Alexa647 secondary antibodies and mounted. Samples were imaged on a Leica SP8 microscope adapted with a HCX PL 63× 1.32 oil objective, solid-state lasers, and HyD detectors. Colocalisation was reported as Mander's coefficient calculated using JACoP plug-in for ImageJ on the basis of two independent experiments.

**LDL and EGF trafficking assays.** For LDL trafficking assays, cells were starved 30 min in RPMI with 5% LPDS at 37 °C, labelled 15 min at 4 °C with 20 µg/ml Dil-LDL in RPMI with 5% LPDS and 10 mM HEPES, washed and chased for indicated times in full medium. For EGF trafficking, cells were starved for 1 h in HBSS (Gibco), incubated with 100 ng/ml EGF-AlexaFluor 555 for 5 min at 4 °C, washed and chased in full medium. At indicated times, cells were fixed in 4% formaldehyde and either mounted directly or stained for indicated markers as described above. Fluorescent staining was recorded using a Zeiss LSM880 Confocal microscope with a 63× oil objective. EEA1+ and LAMP1+ vesicles were annotated in an automated fashion using Volocity v6.3 software (Perkin-Elmer) and the percentage EGF within these vesicles was determined as a percentage of total EGF fluorescence for 5–6 fields per sample with >8 cells per field. To include luminal content of enlarged LAMP1+ vesicles area filling function was enabled and enlarged vesicles were trimmed back by one pickle to prevent artificial area merging in areas of high vesicle density. Statistical significance was determined using two-tailed Student's

T-Test with **p < 0.01 and ***p < 0.001 and error bars reflecting standard error of mean (SEM).

**Lysosomal cholesterol export assay.** Cells were seeded on coverslips and treated for 24 h with 2 µM U18666A (Santa Cruz) in RPMI with 10% FCS, washed with PBS and either fixed directly with 4% paraformaldehyde (pulse); or incubated for 24 h in RPMI with 5% LPDS, 10 µM mevastatin and 50 µM mevalonate (chase), then fixed. Coverslips were stained with Filipin and anti-CD63 antibody and analysed using a Zeiss LSM880 Confocal microscope with a 63× oil objective. Pearson's correlation between Filipin and CD63 staining was determined for three independent experiment using Volocity software (Perkin-Elmer) with 6 fields per sample and >8 cells per field. Statistical significance was determined using two-tailed Student's T-Test with **p < 0.01 and ***p < 0.001 and error bars reflecting standard error of mean (SEM).

**Fluid phase endocytosis.** Bulk endocytosis and delivery of extracellular materials to the late endosomal compartment was measured using the dye Sulforhodamine 101 (SR101, Sigma), following published protocols[44,72]. Briefly, Cells were seeded in live cell imaging dishes (MatTek), preincubated with Lysotracker Green (Molecular Probes) for 30 min, and SR101 dye was added directly to the cell medium at 25 µg/ml. Samples were monitored by time-lapse on a Leica SP8 microscope adapted with a climate control chamber using a HCX PL 63× 1.32 oil objective and images taken at multiple positions at 2 min intervals. SR101 colo-calisation with the Lysotracker-positive compartment is reported as Mander's coefficient calculated using JACoP plug-in for ImageJ on the basis of two independent experiments. Total SR101 entering both cell lines was quantified at indicated time points by flow cytometry. Error bars indicate SD of the mean. Statistical evaluations report on Student's T Test (analysis of two groups), with *p < 0.05, **p < 0.01, ***p < 0.001.

**Electron microscopy.** Samples were prepared for cryo sectioning as follows[73,74]. C18orf8-deficient cells and wild-type cells were fixed for either 2 h in freshly prepared 2% paraformaldehyde and 0.2% glutaraldehyde in 0.1 M phosphate buffer, or for 24 h in freshly prepared 2% paraformaldehyde in 0.1 M phosphate buffer. Fixed cells were scraped, embedded in 12% gelatine (type A, bloom 300, Sigma) and cut with a razor blade into 0.5 mm³ cubes. The sample blocks were infiltrated in phosphate buffer containing 2.3 M sucrose. Sucrose-infiltrated sample blocks were mounted on aluminium pins and plunged in liquid nitrogen. The vitrified samples were stored under liquid nitrogen.

The frozen sample was mounted in a cryo-ultramicrotome (Leica). The sample was trimmed to yield a squared block with a front face of about 300 × 250 µm (Diatome trimming tool). Using a diamond knife (Diatome) and antistatic devise (Leica) a ribbon of 75 nm thick sections was produced that was retrieved from the cryo-chamber with the lift-up hinge method[75]. A droplet of 1.15 M sucrose was used for section retrieval. Obtained sections were transferred to a specimen grid previously coated with formvar and carbon.

Grids containing thawed cryo-sections of cells fixed with 2% paraformaldehyde and 0.2% glutaraldehyde were incubated on the surface of 2% gelatine at 37 °C. Subsequently grids were rinsed to remove the gelatine and sucrose and were embedded in 1.8% methylcellulose and 0.6% uranyl acetate. In case of additional gold-labelling, sections of cells fixed with 2% paraformaldehyde were incubated on drops with 1 µM TNM-BODIPY for 30 min on ice, washed, blocked with 1% BSA in PBS, and then incubated with rabbit polyclonal anti-BODPIY antibody[47]. Sections were subsequently labelled with 10 nm protein A-coated gold particles (CMC, Utrecht University). EM imaging was performed with a Tecnai 20 transmission electron microscope (FEI) operated at 120 kV acceleration voltage.

**Identification of C18orf8 and NPC1 interaction partners by mass spectrometry (MS).** HeLa cells, or HeLa cells expressing 3xHA-C18orf8, C18orf8-3xHA, GFP-3xHA or NPC1-3xHA were lysed in 1% Digitonin. HA-tagged proteins were immune precipitated for 2.5 h at 4 °C using Ezview anti-HA agarose (Sigma-Aldrich) as described above, eluted for 1 h at 37 °C using 0.5 µg/ml HA peptide (Sigma-Aldrich) and denatured using 2% SDS, 50 mM Tris pH7.4. Eluted samples were reduced with 10 mM TCEP for 10 mins RT and alkylated with 40 mM Iodoacetamide for 20 mins RT in the dark. Reduced and alkylated samples were then submitted to digestion using the SP3 method (PMID: 25358341). Briefly, carboxylate coated paramagnetic beads are added to the sample and protein is bound to the beads by acidification with formic acid and addition of acetonitrile (ACN, final 50%). The beads are then washed sequentially with 100% ACN, 70% Ethanol (twice) and 100% ACN. In all, 10 µL of a buffer of TEAB (Triethy-lammonium bicarbonate) pH 8 and 0.1% Sodium deoxycholate (SDC) is then added to the washed beads along with 100 ng trypsin. Samples were then incubated overnight at 37 °C with periodic shaking at 2000 rpm. After digestion, peptides are immobilised on beads by addition of 200 µL ACN and washed twice with 100 µL ACN before eluting in 19 µL 2% DMSO and removing the eluted peptide from the beads.

**MS data acquisition.** Samples were acidified by addition of 1 µL 10% TFA and the whole, 20 µL sample injected. Data were acquired on an Orbitrap Fusion mass

spectrometer (Thermo Scientific) running Xcalibur v4.3 (Thermo-Fisher) coupled to an Ultimate 3000 RSLC nano UHPLC (Thermo Scientific). Samples were loaded at 10 μl/min for 5 min on to an Acclaim PepMap C18 cartridge trap column (300 μm × 5 mm, 5 μm particle size) in 0.1% TFA. After loading a linear gradient of 3–32% solvent B over 60 or 90 min was used for sample separation with a column of the same stationary phase (75 μm × 75 cm, 2 μm particle size) before washing at 90% B and re-equilibration. Solvents were A: 0.1% FA and B:ACN/0.1% FA. MS settings were as follows. MS1: Quadrupole isolation, 120,000 resolution, 5e5 AGC target, 50 ms maximum injection time, ions injected for all parallelisable time. MS2: Quadrupole isolation at an isolation width of m/z 0.7, HCD fragmentation (NCE 34) with the ion trap scanning out in rapid mode from, 8e3 AGC target, 250 ms maximum injection time, ions accumulated for all parallelisable time. Target cycle time was 2 s.

**MS data analysis**. Spectra were searched by Mascot v2.3 (Matrix Science) within Proteome Discoverer 2.1 (Thermo-Fisher) in two rounds of searching. First search was against the UniProt Human reference proteome (26/09/17) and compendium of common contaminants (GPM). The second search took all unmatched spectra from the first search and searched against the human trEMBL database (UniProt, 26/0917). The following search parameters were used. MS1 Tol: 10 ppm, MS2 Tol: 0.6 Da, Fixed mods: Carbamindomethyl (C) Var mods: Oxidation (M), Enzyme: Trypsin (/P). PSM FDR was calculated using Mascot percolator and was controlled at 0.01% for 'high' confidence PSMs and 0.05% for 'medium' confidence PSMs. Proteins were quantified using the Minora feature detector within Proteome Discoverer and values normalised against the median protein abundance of the whole sample.

**Statistics and reproducibility**. The number of replicates of key experiments is indicated in the figure legends and/or methods section. Where not specifically indicated, experiments were performed once. All biological and biochemical experiments were performed with appropriate internal negative and/or positive controls as indicated. Most results were validated by different approaches and/or using alternative techniques as extensively reported in the manuscript. Confocal microscopy images shown are representative of at least 5 fields, with co-localisation quantified as indicated in the figure legends and methods section.

## Data availability

The authors declare that all data supporting the findings of this study are available within the paper and its supplementary information files. Source data are provided with this paper. The mass spectrometry proteomics data have been deposited to the ProteomeXchange Consortium via the PRIDE partner repository[76] with the dataset identifier PXD021444.

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

## Acknowledgements

We are grateful to the following people for their help in this study: Dr. Michael Bassik (Stanford University) for kind donation of the genome-wide CRISPR/Cas9 sgRNA library and Prof. Ron Kopito and Dr. Matthew Porteus (Stanford University) for the pDonor CRISPR knock-in plasmid. FACS experiments were enabled by Dr. Reinard Schulte, Dr. Anna Petrunkina Harrison, the CIMR and JCBC FACS core facilities and confocal microscopy by Matthew Gratian and Mark Bowen. Stuart Bloor kindly assisted with Illumina sequencing, Dr. Ildar Gabaev with design of the top1000 CRISPR library, Dr. Peter Bailey with data analysis and Dr. James Williamson analyzed samples by mass spectrometry. Special thanks to Dr. Zhengzheng Sophia Liang (Harvard University) for fruitful discussions and moral support. This work was financially supported by a Wellcome Trust Principal Research Fellowship to P.J.L. (084957/Z/08/Z). J.P.L. was supported by MRC research grant MR/R0009015/1 and JJCN by ERC grant ERCOPE (GA 694307). The Cambridge Institute for Medical Research (CIMR) was in receipt of a Wellcome Trust strategic award (100140).

## Author contributions

D.J.H.B. and P.J.L. conceived the project. Experiments were carried out by D.J.H.B., A.S., I.B., M.L.M.J. and D.M.E. DJHB analysed the data and prepared the figures. D.J.H.B. and P.J.L. wrote the paper. J.J.C.N. and J.P.L. advised on the project and critically reviewed the paper.

## Competing interests

The authors declare no competing interests.
