## [Peer Review File · Nature Communications]

Reviewers' comments:

Reviewer #1 (Remarks to the Author):

The manuscript „A trimeric Rab7 GEF controls NPC1-dependent lysosomal cholesterol export“ by van den Boomen et al contributes significantly to understanding the mechanism and regulation of lysosomal cholesterol export.

Maintaining cellular cholesterol balance is crucial for eukaryotic cells. In order to regulate the cellular level of cholesterol, cells tightly regulate its exogenous uptake and endogenous biosynthesis. The authors were interested in identifying proteins involved in the maintenance of cellular cholesterol levels. Using an unbiased screening approach, they identified the previously unknown factor C18orf8 as an important protein involved in lysosomal cholesterol export. The authors were able to demonstrate that there is a mechanistic link between the protein C18orf8, its binding to the Rab7-activator Mon1-Ccz1, Rab7 activation, and Rab7-binding to the cholesterol transporter NPC1. The authors use a combination of screening, biochemical characterizations, and cellular assays to establish this link.

The manuscript is well written and the conclusions are fully supported by the experiments. The authors present a convincing study in which they reveal several new findings: A) A new integral component (i.e. C18orf8) of the Rab7-activator Mon1-Ccz1 has been identified, constituting the heterocomplex MCC. B) MCC activates Rab7. C) Activated Rab7 binds NPC1. D) Rab7-binding regulates NPC-activity. The authors discuss their findings with respect to current literature in an exemplary manner.

The wealth and high quality of data presented in this manuscript make it a suitable candidate for publication. I do not have any reservations that this work will be of general interest.

Minor comments

- The authors may want to consider referring to C18orf8 as RMC1 (regulator of Mon1 Ccz1) as suggested in a recent publication. Additionally, they may want reconsidering the name of the Rab7-GEF (MCR instead of MCC). This is up to the author's discretion. However, I personally would appreciate avoiding two different names for one protein (or protein complex).
- Please label all y-axes appropriately.
- Please provide references to Phyre and HHPred.
- line 231: Please indicate the cells used for the pull-downs.
- Related to Figure 5A and 8A: Please also provide a proper list of the MS-protein-identifications as a separate table in the supplement. All proteins identified (likely more than shown in Figures 5A and 8A) should be listed.

Reviewer #2 (Remarks to the Author):

NCOMMS-18-27310A-Z
Van den Boomen et al.

In this manuscript the authors report results of a genome-wide CRISPR screen using and HMGS reporter to identify genes involved in regulation of cholesterol homeostasis. Among the >70 genes

identified by the screen, they found a role for C18orf8, a core component of the Mon1-Ccz1 GEF for Rab7. C18orf8-deficient cells fail to activate Rab7 and accumulate lysosomal cholesterol similar to the NPC1 phenotype. They further show that Rab7 interacts with NPC1 and that absence of the GEF complex components phenocopying NPC1-deficiency. The authors conclude that the trimeric GEF complex plays a central role in Rab7-mediated cholesterol export from lysosomes.

This study sheds new light on the post-lysosomal cholesterol export pathway and represents an advance for the field. The CRISPR screen was well-executed and identified a number of new genes involved in cholesterol homeostasis, as well as anticipated genes such as LDLR, NPC1 and NPC2, providing nice validation. The focus on C18orf8 provides further insight specifically into the regulation of Rab7 in cholesterol export, extending previous observations by Pagano and colleagues. The data quality is excellent, and conclusions generally justified by the data presented. Weaknesses are few and mostly relate to need for additional controls and being more circumspect with respect to some claims.

Specific points:

1. Although the screen was successful, the claim that it provided a comprehensive overview of LDL cholesterol uptake and trafficking seems to be an overstatement and does not acknowledge potential limitations in the screen. To what extent did the screen's bias towards genes involved in LDL uptake limit, for example, identification of genes that participate in cholesterol export from the lysosome? Note that the gene identified by the screen is not directly involved in cholesterol exit from the lysosome, but a regulator of Rab7, which has already been implicated in this pathway (ref 64). If the goal was to look at that part of the LDL cholesterol trafficking pathway more focused screens could have been conducted that would have provided greater depth. The authors should be more circumspect in discussing the capabilities of the screen and address the limitations.

2. Data in Fig 4 indicates that C18orf8-deficiency impairs not only endocytic trafficking LE/Ly, but from EE to late endocytic compartments (panel D). Does this imply that Rab7 has a role in early to late trafficking or does the C18orf8-containing GEF regulate other Rabs such as Rab5?

3. Related to point #2 is that Rab9 has also previously been shown to participate in post-lysosomal cholesterol trafficking. Is there any indication that the trimeric GEF also regulates Rab9?

4. Fig 7D lacks a WT control for the TNM staining.

5. The Rab7-NPC1 co-IPs are performed only with overexpressed protein (for the epitope tag IP). The conclusion that Rab7 and NPC1 co-IP would be strengthened if the reciprocal IP were shown for endogenous proteins.

Minor:

"Remarkably" appears three times in the manuscript. Consider limiting usage for better emphasis.

Reviewer #3 (Remarks to the Author):

This submission by van den Boomen and colleagues focuses on the identification of new genes involved in cell intrinsic cholesterol homeostasis. To accomplish this, they first establish a reporter system in HeLa cells, in which they engineer a fluorescent marker into the C'terminus of the gene HMGCS1, which is regulated by the transcription factor SREBP2. They then conduct a flow cytometry screen using a genome-wide CRISPR library to identify genes whose loss leads to SREBP2-dependent transcription, which in this assay means cells that up-regulate the fluorescent marker even when cholesterol is still available in the media. They identify numerous positive control genes such as LDLR

and NPC1 as top hits, which provides confidence that the assay is working. Following the screen, they mostly focus on a novel gene that scored, C18orf8. They validate the phenotype and, nicely, perform gold-standard cDNA rescue experiments. They further characterize this gene by both proteomic and imaging approaches, placing it in a complex with two other gene products that function as a GEF to regulate Rab7 activity.

This is a straightforward study with a clear conclusion. The manuscript is generally well-written and the experimental logic is easy to follow. We provide some suggestions to improve the clarity and usefulness of the data that were generated. As the novelty of this manuscript may be limited -- as the authors note, another group (ref. 52) recently characterized C18orf8 -- it is especially important that the screen itself serve as a true resource.

Major points

1) Although many positive control genes are identified in the screen, it is important that the authors respect the limitations of a single screen -- no replicates were performed -- conducted in one cell line. Thus, some statements need to be clarified to reflect this reality, such as (line 161): "Our cholesterol reporter screen therefore provides a comprehensive overview and unique insight into known and unknown factors essential for mammalian LDL-cholesterol import..." Indeed, we really have no sense of the false negative rate of this screen, but it may be substantial. For example, based on the results presented in Figure 5, it seems reasonable to expect that Mon1A/B and Ccz1 should also have scored in the primary screen, but they did not. There are several other places where there are claims that this screen is comprehensive, but this is clearly inaccurate.

2) In the supplementary Table S1, the data are only presented as "LogP" and "Score" values. It is crucial to include all raw data from a screen. The standard in the field is to provide read counts -- for each guide, the number of sequencing reads -- such that people may easily re-analyze the data using their preferred analysis approach. Related, we have some general concerns with the RSA algorithm (and we are not alone!) and thus would encourage the authors to also analyze the screen via a different approach, simply for the sake of comparison. Additionally for analysis of the screen, it does not appear that negative control guides were used in any meaningful way. Seeing how they behave is an important measure of the technical noise in the screen, and would be a nice addition to one or more figures. .

3) A smaller-scale pooled screen of putative hits would greatly improve the study. Since the original screen was not performed in replicate, a secondary small screen would help rule out false-positives, which are especially frequent in flow-based screens. This would also be a relatively simple way of assaying these genes in more than one cell type, which is currently a limitation of the generalizability of these results. Related, in line 693, the authors mention that "sgRNA abundance in the sorted sample was compared against a library sample of similar age." This could introduce a great deal of bias in the results of this screen. Why was an unsorted sample not collected in parallel? Conducting a secondary screen with appropriately matched control samples would mitigate this concern.

Minor points

1) With regards to the flow cytometry assay after library infection, please provide specific cell counts for both the pre- and post-sort population in both rounds of sorting. In other words, how many cells underwent FACS, and how many were collected in the positive bin?

2) In Figure 2A, the top hits are grouped into specific biological pathways. How were these genes annotated? Is there a specific annotation source, or was this done manually?

3) Methods: how much gDNA was used for post-screen processing?

4) Methods: what was the source of Cre recombinase?

5) We believe there is a typo in Figure 4 legend - should be leupeptin/E-64D/pepstatin.

Response to reviewers' questions:

Editorial:

Notably, we require that you investigate whether the trimeric GEF regulates other Rab proteins as well as showing the endogenous interaction between Rab7 and NPC1 (see Reviewer #2 comments).

Response: To address these concerns we examined a putative interaction between the trimeric Mon1-Ccz1-C18orf8 GEF and other endosomal GTPases (Rab5, Rab9, Rab11, Arl8B). We find the trimeric MCC GEF selectively binds to a dominant-negative Rab7 (T22N) and none of the other GTPases (Fig S7A). Furthermore, we show that the upregulation of our HMGCS1-Clover reporter in C18orf8-deficient cells can be restored by a constitutively active Rab7 (Q67L), but not constitutively active mutants of Rab5, Rab9, Rab11 or Arl8B (Fig S7B). The MCC GEF therefore appears to specifically regulate Rab7 and C18orf8-deficiency defects are due to a specific lack of Rab7 activation.

The interaction between endogenous Rab7 and NPC1 is shown by reciprocal immune precipitation in Fig 8C (NPC1 immune precipitation) and Fig 8D (Rab7 immune precipitation).

Since, from an editorial point of view, we are particularly interested in the screen as a resource for the research community, we require that you address the concerns of Reviewers #2 and #3 pertaining to the screen with care. Importantly, please tone down the claims of the comprehensive nature of the screen where appropriate, show the raw data from the primary screen, and perform a smaller-scale pooled screen of putative hits to make your findings more robust and generalizable.

Response: We have toned down our reference to the comprehensive nature of the screen and refer to it as extensive to indicate the broad range of hits without a claim of completeness.

We have performed a smaller-scale pooled screen using a newly generated sgRNA library composed of sgRNAs for the top 1000 scoring genes in our genome-wide CRISPR screen, as suggested by reviewer 3 (Fig S2A). Results of this targeted screen show wide-range overlap with that of our genome-wide screen (Fig S2B) - thus emphasising the reproducibility of our approach. The new hits obtained are largely complementary to the previous hits, thus creating an extensive dataset that is an important resource for future research. All raw data has been included in Table S1; an overview of combined hits involved in membrane trafficking has been included Fig 2C.

Reviewer 1:

The wealth and high quality of data presented in this manuscript make it a suitable candidate for publication. I do not have any reservations that this work will be of general interest.

Minor comments

- The authors may want to consider referring to C18orf8 as RMC1 (regulator of Mon1 Ccz1) as suggested in a recent publication. Additionally, they may want reconsidering the name of the Rab7-GEF (MCR instead of MCC). This is up to the author's discretion. However, I personally would appreciate avoiding two different names for one protein (or protein complex).

Response: We understand the importance of naming, however we feel the name 'regulator of Mon1-Ccz1' (RMC) is misleading. This name was given largely based on C18orf8 binding to Ccz1-Mon1, without much functional evidence of it acting as a regulator. We find C18orf8 is not in fact a regulator of this GEF complex, but an intrinsic subunit of the complex. We therefore prefer to avoid the name RMC1 and refer to the protein by its original name C18orf8.

- Please label all y-axes appropriately.

- Please provide references to Phyre and HHPred.

- line 231: Please indicate the cells used for the pull-downs.

Response: The requested changes have been made. Flow cytometry plots have been annotated on the vertical axis. References to Phyre2 (40) and HHPred (41) have been added. Line 259 (former 231) has been updated to indicate pull-downs were performed in HeLa cells.

- Related to Figure 5A and 8A: Please also provide a proper list of the MS-protein-identifications as a separate table in the supplement. All proteins identified (likely more than shown in Figures 5A and 8A) should be listed.

Response: A complete list of proteins identified by MS in C18orf8 and NPC1 immune precipitations has been added in Table S2.

Reviewer 2:

This study sheds new light on the post-lysosomal cholesterol export pathway and represents an advance for the field. The CRISPR screen was well-executed and identified a number of new genes involved in cholesterol homeostasis, as well as anticipated genes such as LDLR, NPC1 and NPC2, providing nice validation. The focus on C18orf8 provides further insight specifically into the regulation of Rab7 in cholesterol export, extending previous observations by Pagano and colleagues. The data quality is excellent, and conclusions generally justified by the data presented. Weaknesses are few and mostly relate to need for additional controls and being more circumspect with respect to some claims.

Specific points:

1. Although the screen was successful, the claim that it provided a comprehensive overview of LDL cholesterol uptake and trafficking seems to be an overstatement and does not acknowledge potential limitations in the screen. To what extent did the screen's bias towards genes involved in LDL uptake limit, for example, identification of genes that participate in cholesterol export from the lysosome? Note that the gene identified by the screen is not directly involved in cholesterol exit from the lysosome, but a regulator of Rab7, which has already been implicated in this pathway (ref 64). If the goal was to look at that part of the LDL cholesterol trafficking pathway more focused screens could have been conducted that would have provided greater depth. The authors should be more circumspect in discussing the capabilities of the screen and address the limitations.

Response: We agree that our screening results are extensive but not complete and apologize if this was not clear from the manuscript. We have removed all reference to the screens comprehensive nature, but rather use the word extensive. Addition of a second targeted screen (see reviewer 3, major point 3) further adds to the range of genes covered. The discussion has been updated to reflect the consideration 'As no single genetic screening approach is sufficient to give a complete picture of a complex cellular pathway, our two screens are complementary as well as overlapping, with multi-subunit complexes identified across screens' (line 445).

The goal of our study was to identify genes required for cellular cholesterol homeostasis (as stated at the outset of our results section), not to specifically screen for genes involved in lysosomal cholesterol export. We are not aware of a bias in our screening approach towards LDL uptake versus lysosomal cholesterol export. Our screen hits a large number of genes in the early secretory pathway, endocytosis and endosomal trafficking, yet components of the lysosomal cholesterol export machinery, noticeably NPC1, NPC2 and Rab7, are also major hits. Putative lipid transfer proteins

down-stream of NPC1 might show a high level of redundancy (see below), which might explain their absence from the screen.

*The reviewer is correct that C18orf8 itself is not directly involved in NPC1 -dependent lysosomal cholesterol export, but is indirectly involved via its activation of Rab7. This has been emphasised within the manuscript and is clearly depicted in our schematic in **Fig 10**.*

It is important to point out that, although overexpression of exogenous Rab7 was previously shown to complement cholesterol and sphingolipid accumulation in NPC1 mutant patient fibroblasts (Choudhury et al. 2002), this is not the same as demonstrating the direct involvement of Rab7 in lysosomal cholesterol export. It suggests that its overexpression can complement for an existing cholesterol export defect.

The mechanism behind this remarkable effect remained unclear and was previously suggested to involve enhanced vesicular trafficking to the Golgi apparatus. We show Rab7 interacts with the NPC1 cholesterol transporter and this interaction is essential for direct NPC1-dependent cholesterol export from lysosomes. As such we add mechanism to an existing observation.

Although multiple lipid transfer proteins have been implicated in lysosomal cholesterol export, in our experience their depletion typically yields only minor cholesterol defects and the process as a whole remains enigmatic. We find that a defect in Rab7 activation leads to a full Niemann-Pick-like phenotype in wild-type cells, thus placing Rab7 and the MCC GEF at the centre of lysosomal cholesterol export.

2. Data in Fig 4 indicates that C18orf8-deficiency impairs not only endocytic trafficking LE/Ly, but from EE to late endocytic compartments (panel D). Does this imply that Rab7 has a role in early to late trafficking or does the C18orf8-containing GEF regulate other Rabs such as Rab5?

*Response: This is an important point. The dimeric Mon1-Ccz1 complex was previously characterized in yeast and Drosophila where it has been shown to specifically activate Rab7, which drives early-to-late endosomal trafficking (Poteryaev et al. 2010; Kinchen and Ravichandran 2010; Nordmann et al. 2010; Langemeyer et al. 2020). Our observations concur with this premise. To assess whether the mammalian Mon1-Ccz1-C18orf8 (MCC) complex shows specificity for Rab7 or might indeed affect Rab5 or Rab9, we assessed the interaction between MCC and a range of endocytic GTPases, including Rab5, Rab7, Rab9, Rab11 and Arl8B. The results in a new **Fig S7A** show MCC interacts only with Rab7.*

*To assess whether the endocytic defects observed in C18orf8-deficient cells are caused solely by a defect in Rab7 activation, we complemented C18orf8-deficient cells with a constitutively active Rab5, Rab7, Rab9, Rab11 and Arl8B. We find only a constitutively active Rab7 is able to restore HMGCS1-Clover expression (and hence lysosomal cholesterol export) to wild-type levels in C18orf8-deficient cells, whereas a constitutively active Rab5, Rab11 and Arl8B aggravate the C18orf8-deficiency phenotype and a constitutively active Rab9 has no effect (**Fig S7B**). Defects in C18orf8-deficient cell are thus caused specifically by an inability to activate Rab7.*

3. Related to point #2 is that Rab9 has also previously been shown to participate in post-lysosomal cholesterol trafficking. Is there any indication that the trimeric GEF also regulates Rab9?

Response: see above at point 2. We have no evidence to suggest the MCC GEF regulates Rab9; our data suggest high specificity for Rab7. The partial restoration of cholesterol accumulation by Rab9 overexpression in NP-C fibroblasts and mice (Choudhury et al. 2002; Kaptzan et al. 2009) might reflect enhanced vesicular recycling. This goes beyond the scope of this manuscript in which we focus on the MCC GEF, Rab7 and direct lysosomal cholesterol export.

4. Fig 7D lacks a WT control for the TNM staining.

*Response: We apologize for this. Due to the ongoing Covid-19 pandemic we unfortunately have no access to electron microscopy. Wild-type controls are available for all Filipin free cholesterol stainings in **Fig 7A,C,E**. TNM staining is a well-established method (Edgar et al. 2016; Nishimura et al. 2013) and used here merely to exemplify endosomal cholesterol accumulation at the ultrastructural level. However, if the reviewer prefers, we would consider removing **Fig 7D** as it is not essential to the story.*

5. The Rab7-NPC1 co-IPs are performed only with overexpressed protein (for the epitope tag IP). The conclusion that Rab7 and NPC1 co-IP would be strengthened if the reciprocal IP were shown for endogenous proteins.

*Response: Thank you for this suggestion. The manuscript has been updated to show the interaction between endogenous Rab7 and NPC1 using reciprocal immune precipitation of NPC1 (**Fig 8D**) and Rab7 (**Fig 8E**) with NPC1 and Rab7 knock-out cells as controls.*

Reviewer 3:

This is a straightforward study with a clear conclusion. The manuscript is generally well-written and the experimental logic is easy to follow. We provide some suggestions to improve the clarity and usefulness of the data that were generated. As the novelty of this manuscript may be limited -- as the authors note, another group (ref. 52) recently characterized C18orf8 -- it is especially important that the screen itself serve as a true resource.

Major points

1) Although many positive control genes are identified in the screen, it is important that the authors respect the limitations of a single screen -- no replicates were performed -- conducted in one cell line. Thus, some statements need to be clarified to reflect this reality, such as (line 161): “Our cholesterol reporter screen therefore provides a comprehensive overview and unique insight into known and unknown factors essential for mammalian LDL-cholesterol import...” Indeed, we really have no sense of the false negative rate of this screen, but it may be substantial. For example, based on the results presented in Figure 5, it seems reasonable to expect that Mon1A/B and Ccz1 should also have scored in the primary screen, but they did not. There are several other places where there are claims that this screen is comprehensive, but this is clearly inaccurate.

Response: We are aware that any one genetic screen is unlikely to show a complete picture of all genes involved in a certain pathway and apologize if this was unclear from the manuscript. The discussion has been updated to reflect this, indicating ‘As no single genetic screening approach is sufficient to give a complete picture of a complex cellular pathway, our two screens are complementary as well as overlapping, with multi-subunit complexes identified across screens’ (line 445). We have removed all reference to the screens comprehensive nature, but rather use the word extensive. Reproducibility of the screen is addressed under point 3.

*The absence of Mon1A/B and Ccz1 from our screening results has concerned us for a while as these proteins are clearly of great importance to our paper. Mon1A and Mon1B show cellular redundancy and need to be depleted in combination to obtain an HMGCS1-Clover^{high} phenotype (**Fig 4E**), which explains their absence in our genetic screen. The absence of Ccz1 was more puzzling as its depletion clearly gives a robust phenotype with little toxicity (**Fig 4E**).*

Looking further into this issue we realized Ccz1 is encoded by two near-identical genes, Ccz1 and Ccz1b on chromosome 7, that yield identical proteins. Both Ccz1 and Ccz1b are present in the Bassik genome-wide CRISPR library (Morgens et al. 2017) applied here. Inspection of their respective sgRNAs unfortunately shows these are duplicated between the two genes, reflecting a slightly imperfect design of the CRISPR library (which we did not generate).

*During our initial analysis, reads from duplicate sgRNAs were removed as default Bowtie settings only allow read alignment to a single sgRNA. Modification of this protocol to allow alignment of a single read to two sgRNAs allows Ccz1 to become a significant hit in our screen. Other genes that encountered a similar problem include VPS33A, Rab5C, COG8 and TMEM251 (these genes share sgRNAs with overlapping lncRNAs). **Fig 1G** and the Methods section have been updated to reflect our new analysis. Identical sgRNAs were avoided during creation of the top1000 library (see below).*

There is clear novelty in the manuscript. Although C18orf8 was recently identified as an interaction partner of mammalian Mon1-Ccz1, its characterisation is very limited. We characterise C18orf8 as an integral subunit, not a regulator, of the mammalian trimeric Mon1-Ccz1-C18orf8 complex, essential for complex stability and function. We also demonstrate that all three MCC subunits are essential for the activation of mammalian Rab7, creating similar deficiency phenotypes and marking MCC as the sole Rab7 GEF in mammalian cells.

Beyond Rab7 activation, our study yields significant mechanistic insight into lysosomal cholesterol export. Although several components of membrane contact sites are reported to be required for NPC1-dependent cholesterol transfer, many of these studies suffer from weak, incomplete phenotypes and are somewhat contradictory. We demonstrate that Rab7 interacts with the NPC1 cholesterol transporter in an activation-dependent manner and that deficient Rab7 activation causes a complete block in lysosomal cholesterol export and a Niemann-Pick-like phenotype. Together these findings place Rab7 at the centre of this important process.

2) In the supplementary Table S1, the data are only presented as “LogP” and “Score” values. It is crucial to include all raw data from a screen. The standard in the field is to provide read counts -- for each guide, the number of sequencing reads -- such that people may easily re-analyze the data using their preferred analysis approach. Related, we have some general concerns with the RSA algorithm (and we are not alone!) and thus would encourage the authors to also analyze the screen via a different approach, simply for the sake of comparison. Additionally for analysis of the screen, it does not appear that negative control guides were used in any meaningful way. Seeing how they behave is an important measure of the technical noise in the screen, and would be a nice addition to one or more figures.

*Response: Full CRISPR screening results, including read-counts of both screens, have been added to Table S1. We have reanalysed our CRISPR screens (genome-wide and targeted) using the MAGeCK algorithm. The results look almost identical. To reflect the reviewers concerns about RSA we have removed RSA analysis from our results and updated figures to reflect the new MAGeCK analysis. A figure reflecting negative control guides has been added as **Fig S10** and is referenced in line 695.*

3) A smaller-scale pooled screen of putative hits would greatly improve the study. Since the original screen was not performed in replicate, a secondary small screen would help rule out false-positives, which are especially frequent in flow-based screens. This would also be a relatively simple way of assaying these genes in more than one cell type, which is currently a limitation of the generalizability of these results. Related, in line 693, the authors mention that “sgRNA abundance in the sorted sample was compared against a library sample of similar age.” This could introduce a great deal of bias in the results of this screen. Why was an unsorted sample not collected in parallel? Conducting a secondary screen with appropriately matched control samples would mitigate this concern.

Response: A targeted smaller screen was performed as suggested. For this purpose we created a targeted sgRNA library comprising the top 1000 rating genes in our genome-wide CRISPR screen (Fig 2A). Screening HMGCS1-Clover using this library we obtained 65 hits (Fig S2). Of these 65 hits, 52 overlapped with our genome-wide screen (Fig 2B), thus proving high reproducibility of our screening approach. Of the hits specific to either screen alone, many are involved in membrane trafficking with subunits of functional membrane trafficking complexes often present in either screen alone or both. The two datasets are thus overlapping and complementary.

The primary/secondary screening approach has thus validated as well as extended our previous screening data and created an extensive dataset that serves as a useful resource for future research. We appreciate this suggestion of the reviewer.

A new Fig S2 shows the results of our targeted CRISPR screen; its comparison with the genome-wide screen is depicted in Fig 2B; the discussion and Fig 2C have also been updated to reflect the new data. A matched library sample was harvested at the same as the sorted sample in the new targeted CRISPR screen and the raw data is available in Table S1. We are currently unable to screen the HMGCS1-Clover in cell lines other than HeLa as this would require the generation of a new knock-in cell line, which not practical at present. As a house-keeping function, the LDL uptake pathway should be highly conserved between cell types.

Minor points

1) With regards to the flow cytometry assay after library infection, please provide specific cell counts for both the pre- and post-sort population in both rounds of sorting. In other words, how many cells underwent FACS, and how many were collected in the positive bin?

Response: The Materials and Methods has been updated to reflect cell counts at sorting stages.

2) In Figure 2A, the top hits are grouped into specific biological pathways. How were these genes annotated? Is there a specific annotation source, or was this done manually?

Response: Top hits were grouped manually into biological pathways as indicated in the Materials and Methods section.

3) Methods: how much gDNA was used for post-screen processing?

Response: The Materials and Methods has been updated to include this information. For the genome-wide CRISPR screen 400µg and 100µg gDNA was used respectively for post-screen processing of the library and sorted sample, while for the targeted sub-genomic screen, 200µg and 14µg gDNA was used.

4) Methods: what was the source of Cre recombinase?

Response: The Cre recombinase used here is expressed from a lentiviral plasmid and derives from Bacteriophage P1. The supplemental methods have been updated to reflect this fact.

5) We believe there is a typo in Figure 4 legend - should be leupeptin/E-

64D/pepstatin. *Response: The typo has been corrected.*

References:

- Choudhury, A., M. Dominguez, V. Puri, D. K. Sharma, K. Narita, C. L. Wheatley, D. L. Marks, and R. E. Pagano. 2002. 'Rab proteins mediate Golgi transport of caveola-internalized glycosphingolipids and correct lipid trafficking in Niemann-Pick C cells', *J Clin Invest*, 109: 1541-50.
- Edgar, J. R., P. T. Manna, S. Nishimura, G. Banting, and M. S. Robinson. 2016. 'Tetherin is an exosomal tether', *Elife*, 5.
- Kinchen, J. M., and K. S. Ravichandran. 2010. 'Identification of two evolutionarily conserved genes regulating processing of engulfed apoptotic cells', *Nature*, 464: 778-82.
- Langemeyer, L., A. C. Borchers, E. Herrmann, N. Fullbrunn, Y. Han, A. Perz, K. Auffarth, D. Kummel, and C. Ungermann. 2020. 'A conserved and regulated mechanism drives endosomal Rab transition', *Elife*, 9.
- Morgens, D. W., M. Wainberg, E. A. Boyle, O. Ursu, C. L. Araya, C. K. Tsui, M. S. Haney, G. T. Hess, K. Han, E. E. Jeng, A. Li, M. P. Snyder, W. J. Greenleaf, A. Kundaje, and M. C. Bassik. 2017. 'Genome-scale measurement of off-target activity using Cas9 toxicity in high-throughput screens', *Nat Commun*, 8: 15178.
- Nishimura, S., K. Ishii, K. Iwamoto, Y. Arita, S. Matsunaga, Y. Ohno-Iwashita, S. B. Sato, H. Kakeya, T. Kobayashi, and M. Yoshida. 2013. 'Visualization of sterol-rich membrane domains with fluorescently-labeled theonellamides', *PLoS One*, 8: e83716.
- Nordmann, M., M. Cabrera, A. Perz, C. Brocker, C. Ostrowicz, S. Engelbrecht-Vandre, and C. Ungermann. 2010. 'The Mon1-Ccz1 complex is the GEF of the late endosomal Rab7 homolog Ypt7', *Curr Biol*, 20: 1654-9.
- Poteryaev, D., S. Datta, K. Ackema, M. Zerial, and A. Spang. 2010. 'Identification of the switch in early-to-late endosome transition', *Cell*, 141: 497-508.

REVIEWERS' COMMENTS:

Reviewer #1 (Remarks to the Author):

The authors have addressed all my previous comments to my full satisfaction. Therefore, I am happy to support publication of this work in Nature Communications.

I only suggest putting the name for the supplementary tables S1 and S2 into the spreadsheet. It took me a bit of time figuring out which table the manuscript was referring to due to the cryptic file name.

Reviewer #2 (Remarks to the Author):

Authors have responded constructively to initial review. Since the data in Fig 7D lacks the WT control - and the panel not essential to the overall manuscript - agree with the authors' suggestion of omitting that panel.

Reviewer #3 (Remarks to the Author):

The authors have answered our most pressing concerns, both with additional data as well as the more-appropriate framing of the meaning of the results. We believe this manuscript is ready for publication.

REVIEWERS' COMMENTS:

Reviewer #1 (Remarks to the Author):

The authors have addressed all my previous comments to my full satisfaction. Therefore, I am happy to support publication of this work in Nature Communications.

I only suggest putting the name for the supplementary tables S1 and S2 into the spreadsheet. It took me a bit of time figuring out which table the manuscript was referring to due to the cryptic file name.

Response: *Table S1/S2 have been renamed to Supplementary Data 1/2. As requested, the file name is now indicated on the first tab of each spreadsheet, which also gives a short summary of the file content.*

Reviewer #2 (Remarks to the Author):

Authors have responded constructively to initial review. Since the data in Fig 7D lacks the WT control - and the panel not essential to the overall manuscript - agree with the authors' suggestion of omitting that panel.

Response: *We have inserted a wild-type control for the TNM immuno-gold staining in Fig 7d as initially requested by the reviewer. As the experiment is now fully controlled, we will maintain Fig 7d within the manuscript.*

Reviewer #3 (Remarks to the Author):

The authors have answered our most pressing concerns, both with additional data as well as the more-appropriate framing of the meaning of the results. We believe this manuscript is ready for publication.